# Learning Nearest Neighbor Graphs from Noisy Distance Samples

**Blake Mason** *
University of Wisconsin
Madison, WI 53706
bmason3@wisc.edu

**Ardhendu Tripathy** *
University of Wisconsin
Madison, WI 53706
astripathy@wisc.edu

**Robert Nowak**
University of Wisconsin
Madison, WI 53706
rdnowak@wisc.edu

## Abstract

We consider the problem of learning the *nearest neighbor graph* of a dataset of $n$ items. The metric is unknown, but we can query an oracle to obtain a noisy estimate of the distance between any pair of items. This framework applies to problem domains where one wants to learn people's preferences from responses commonly modeled as noisy distance judgments. In this paper, we propose an active algorithm to find the graph with high probability and analyze its query complexity. In contrast to existing work that forces Euclidean structure, our method is valid for general metrics, assuming only symmetry and the triangle inequality. Furthermore, we demonstrate efficiency of our method empirically and theoretically, needing only $\mathcal{O}(n \log(n) \Delta^{-2})$ queries in favorable settings, where $\Delta^{-2}$ accounts for the effect of noise. Using crowd-sourced data collected for a subset of the UT Zappos50K dataset, we apply our algorithm to learn which shoes people believe are most similar and show that it beats both an active baseline and ordinal embedding.

## 1 Introduction

In modern machine learning applications, we frequently seek to learn proximity/ similarity relationships between a set of items given only noisy access to pairwise distances. For instance, practitioners wishing to estimate internet topology frequently collect one-way-delay measurements to estimate the distance between a pair of hosts [9]. Such measurements are affected by physical constraints as well as server load, and are often noisy. Researchers studying movement in hospitals from WiFi localization data likewise contend with noisy distance measurements due to both temporal variability and varying signal strengths inside the building [4]. Additionally, human judgments are commonly modeled as noisy distances [26, 23]. As an example, Amazon Discover asks customers their preferences about different products and uses this information to recommend new items it believes are similar based on this feedback. We are often primarily interested in the *closest* or *most similar* item to a given one– e.g., the closest server, the closest doctor, the most similar product. The particular item of interest may not be known *a priori*. Internet traffic can fluctuate, different patients may suddenly need attention, and customers may be looking for different products. To handle this, we must learn the closest/ most similar item for *each* item. This paper introduces the problem of learning the *Nearest Neighbor Graph* that connects each item to its nearest neighbor from noisy distance measurements.

---

**Problem Statement:** Consider a set of $n$ points $\mathcal{X} = \{x_1, \cdots, x_n\}$ in a metric space. The metric is unknown, but we can query a stochastic oracle for an estimate of any pairwise distance. In as few queries as possible, we seek to learn a nearest neighbor graph of $\mathcal{X}$ that is correct with probability $1 - \delta$, where each $x_i$ is a vertex and has a directed edge to its nearest neighbor $x_{i^*} \in \mathcal{X} \setminus \{x_i\}$.

## 1.1 Related work

Nearest neighbor problems (from noiseless measurements) are well studied and we direct the reader to [3] for a survey. [6, 30, 25] all provide theory and algorithms to learn the nearest neighbor graph which apply in the noiseless regime. Note that the problem in the noiseless setting is *very* different. If noise corrupts measurements, the methods from the noiseless setting can suffer persistent errors. There has been recent interest in introducing noise via subsampling for a variety of distance problems [24, 1, 2], though the noise here is not actually part of the data but introduced for efficiency. In our algorithm, we use the triangle inequality to get tighter estimates of noisy distances in a process equivalent to the classical Floyd–Warshall [11, 7]. This has strong connections to the metric repair literature [5, 13] where one seeks to alter a set of noisy distance measurements as little as possible to learn a metric satisfying the standard axioms. [27] similarly uses the triangle inequality to bound unknown distances in a related but noiseless setting. In the specific case of noisy distances corresponding to human judgments, a number of algorithms have been proposed to handle related problems, most notably Euclidean embedding techniques, e.g. [17, 31, 23]. To reduce the load on human subjects, several attempts at an active method for learning Euclidean embeddings have been made but have only seen limited success [20]. Among the culprits is the strict and often unrealistic modeling assumption that the metric be Euclidean and low dimensional. In the particular case that the algorithm may query triplets (e.g., "is $i$ or $j$ closer to $k$?") and receive noisy responses, [22] develop an interesting, passive technique under general metrics for learning a relative neighborhood graph which is an undirected relaxation of a nearest neighbor graph.

## 1.2 Main contributions

In this paper we introduce the problem of identifying the *nearest neighbor graph* from noisy distance samples and propose `ANNTri`, an active algorithm, to solve it for general metrics. We empirically and theoretically analyze its complexity to show improved performance over a passive and an active baseline. In favorable settings, such as when the data forms clusters, `ANNTri` needs only $\mathcal{O}(n \log(n)/\Delta^2)$ queries, where $\Delta$ accounts for the effect of noise. Furthermore, we show that `ANNTri` achieves superior performance compared to methods which require much stronger assumptions. We highlight two such examples. In Fig. 2c, for an embedding in $\mathbb{R}^2$, `ANNTri` outperforms the common technique of triangulation that works by estimating each point's distance to a set of anchors. In Fig. 3b, we show that `ANNTri` likewise outperforms Euclidean embedding for predicting which images are most similar from a set of similarity judgments collected on Amazon Mechanical Turk. The rest of the paper is organized as follows. In Section 2, we further setup the problem. In Sections 3 and 4 we present the algorithm and analyze its theoretical properties. In Section 5 we show `ANNTri`'s empirical performance on both simulated and real data. In particular, we highlight its efficiency in learning from human judgments.

## 2 Problem setup and summary of our approach

We denote distances as $d_{i,j}$ where $d : \mathcal{X} \times \mathcal{X} \to \mathbb{R}_{\geq 0}$ is a distance function satisfying the standard axioms and define $x_{i^*} := \arg\min_{x \in \mathcal{X} \setminus \{x_i\}} d(x_i, x)$. Though the distances are unknown, we are able to draw independent samples of its true value according to a stochastic distance oracle, i.e. querying

$$\mathsf{Q}(i, j) \quad \text{yields a realization of} \quad d_{i,j} + \eta, \tag{1}$$

where $\eta$ is a zero-mean subGaussian random variable assumed to have scale parameter $\sigma = 1$. We let $\hat{d}_{i,j}(t)$ denote the empirical mean of the values returned by $\mathsf{Q}(i, j)$ queries made until time $t$. The number of $\mathsf{Q}(i, j)$ queries made until time $t$ is denoted as $T_{i,j}(t)$. A possible approach to obtain the nearest neighbor graph is to repeatedly query all $\binom{n}{2}$ pairs and report $x_{i^*}(t) = \arg\min_{j \neq i} \hat{d}_{i,j}(t)$ for all $i \in [n]$. But since we only wish to learn $x_{i^*} \forall i$, if $d_{i,k} \gg d_{i,i^*}$, we do not need to query $\mathsf{Q}(i, k)$ as many times as $\mathsf{Q}(i, i^*)$. To improve our query efficiency, we could instead adaptively sample to focus queries on distances that we estimate are smaller. A simple adaptive method to find the nearest neighbor graph would be to iterate over $x_1, x_2, \dots, x_n$ and use a best-arm identification algorithm to find $x_{i^*}$ in the $i^{th}$ round.[1] However, this procedure treats each round independently, ignoring properties of metric spaces that allow information to be shared between rounds.

- Due to symmetry, for any $i < j$ the queries $\mathsf{Q}(i,j)$ and $\mathsf{Q}(j,i)$ follow the same law, and we can *reuse* values of $\mathsf{Q}(i,j)$ collected in the $i^{th}$ round while finding $x_{j*}$ in the $j^{th}$ round.

- Using concentration bounds on $d_{i,j}$ and $d_{i,k}$ from samples from $\mathsf{Q}(i,j)$ and $\mathsf{Q}(i,k)$ collected in the $i^{th}$ round, we can bound $d_{j,k}$ via the triangle inequality. As a result, we may be able to state $x_k \neq x_{j*}$ without even querying $\mathsf{Q}(j,k)$.

Our proposed algorithm ANNTri uses all the above ideas to find the nearest neighbor graph of $\mathcal{X}$. For general $\mathcal{X}$, the sample complexity of ANNTri contains a problem-dependent term that involves the order in which the nearest neighbors are found. For an $\mathcal{X}$ consisting of sufficiently well separated clusters, this order-dependence for the sample complexity does not exist.

## 3    Algorithm

Our proposed algorithm (Algorithm 1) ANNTri finds the nearest neighbor graph of $\mathcal{X}$ with probability $1 - \delta$. It iterates over $x_j \in \mathcal{X}$ in order of their subscript index and finds $x_{j*}$ in the $j^{th}$ 'round'. All bounds, counts of samples, and empirical means are stored in $n \times n$ symmetric matrices in order to share information between different rounds. We use Python array/Matlab notation to indicate individual entries in the matrices, for e.g., $\hat{d}[i,j] = \hat{d}_{i,j}(t)$. The number of $\mathsf{Q}(i,j)$ queries made is queried is stored in the $(i,j)^{th}$ entry of $T$. Matrices $U$ and $L$ record upper and lower confidence bounds on $d_{i,j}$. $U^{\triangle}$ and $L^{\triangle}$ record the associated triangle inequality bounds. Symmetry is ensured by updating the $(j,i)^{th}$ entry at the same time as the $(i,j)^{th}$ entry for each of the above matrices. In the $j^{th}$ round, ANNTri finds the correct $x_{j*}$ with probability $1 - \delta/n$ by calling SETri (Algorithm 2), a modification of the successive elimination algorithm for best-arm identification. In contrast to standard successive elimination, at each time step SETri only samples those points in the active set that have the fewest number of samples.

---

**Algorithm 1** ANNTri

---

**Require:** $n$, procedure SETri (Alg. 2), confidence $\delta$
1: Initialize $\hat{d}, T$ as $n \times n$ matrices of zeros, $U, U^{\triangle}$ as $n \times n$ matrices where each entry is $\infty$, $L, L^{\triangle}$ as $n \times n$ matrices where each entry is $-\infty$, NN as a length $n$ array
2: **for** $j = 1$ **to** $n$ **do**
3:     **for** $i = 1$ **to** $n$ **do** {find tightest triangle bounds}
4:         **for all** $k \neq i$ **do**
5:             Set $U^{\triangle}[i,k], U^{\triangle}[k,i], \leftarrow \min_\ell U^{\triangle_\ell}_{i,k}$, see (7)
6:             Set $L^{\triangle}[i,k], L^{\triangle}[k,i] \leftarrow \max_\ell L^{\triangle_\ell}_{i,k}$, see (8)
7:     NN$[j]$ = SETri$(j, \hat{d}, U, U^{\triangle}, L, L^{\triangle}, T, \xi = \delta/n)$
8: **return** The nearest neighbor graph adjacency list NN

---

**Algorithm 2** SETri

---

**Require:** index $j$, callable oracle $\mathsf{Q}(\cdot,\cdot)$ (Eq. (1)), six $n \times n$ matrices: $\hat{d}, U, U^{\triangle}, L, L^{\triangle}, T$, confidence $\xi$
1: Initialize active set $\mathcal{A}_j \leftarrow \{a \neq j : \max\{L[a,j], L^{\triangle}[a,j]\} < \min_k \min\{U[j,k], U^{\triangle}[j,k]\}\}$
2: **while** $|\mathcal{A}_j| > 1$ **do**
3:     **for all** $i \in \mathcal{A}_j$ such that $T[i,j] = \min_{k \in \mathcal{A}_j} T[i,k]$ **do** {only query points with fewest samples}
4:         Update $\hat{d}[i,j], \hat{d}[j,i] \leftarrow (\hat{d}[i,j] \cdot T[i,j] + \mathsf{Q}(i,j))/(T[i,j] + 1)$
5:         Update $T[i,j], T[j,i] \leftarrow T[i,j] + 1$
6:         Update $U[i,j], U[j,i] \leftarrow \hat{d}[i,j] + C_\xi(T[i,j])$
7:         Update $L[i,j], L[j,i] \leftarrow \hat{d}[i,j] - C_\xi(T[i,j])$
8:     Update $\mathcal{A}_j \leftarrow \{a \neq j : \max\{L[a,j], L^{\triangle}[a,j]\} < \min_k \min\{U[j,k], U^{\triangle}[j,k]\}\}$
9: **return** The index $i$ for which $x_i \in \mathcal{A}_j$

---

## 3.1 Confidence bounds on the distances

Using the subGaussian assumption on the noise random process, we can use Hoeffding's inequality and a union bound over time to get the following confidence intervals on the distance $d_{j,k}$:

$$|\hat{d}_{j,k}(t) - d_{j,k}| \leq \sqrt{2\frac{\log(4n^2(T_{j,k}(t))^2/\delta)}{T_{j,k}(t)}} =: C_{\delta/n}(T_{j,k}(t)), \tag{2}$$

which hold for all points $x_k \in \mathcal{X} \setminus \{x_j\}$ at all times $t$ with probability $1 - \delta/n$, i.e.

$$\mathbb{P}(\forall t \in \mathbb{N}, \forall i \neq j, d_{i,j} \in [L_{i,j}(t), U_{i,j}(t)]) \geq 1 - \delta/n, \tag{3}$$

where $L_{i,j}(t) := \hat{d}_{i,j}(t) - C_{\delta/n}(T_{i,j}(t))$ and $U_{i,j}(t) := \hat{d}_{i,j}(t) + C_{\delta/n}(T_{i,j}(t))$. [10] use the above procedure to derive the following upper bound for the number of oracle queries used to find $x_{j^*}$:

$$\mathcal{O}\left(\sum_{k \neq j} \frac{\log(n^2/(\delta \Delta_{j,k}))}{\Delta_{j,k}^2}\right), \tag{4}$$

where for any $x_k \notin \{x_j, x_{j^*}\}$ the suboptimality gap $\Delta_{j,k} := d_{j,k} - d_{j,j^*}$ characterizes how hard it is to rule out $x_k$ from being the nearest neighbor. We also set $\Delta_{j,j^*} := \min_{k \notin \{j,j^*\}} \Delta_{j,k}$. Note that one can use tighter confidence bounds as detailed in [12] and [18] to obtain sharper bounds on the sample complexity of this subroutine.

## 3.2 Computing the triangle bounds and active set $\mathcal{A}_j(t)$

Since $\mathcal{A}_j(\cdot)$ is only computed within SETri, we abuse notation and use its argument $t$ to indicate the time counter private to SETri. Thus, the initial active set computed by SETri when called in the $j^{th}$ round is denoted $\mathcal{A}_j(0)$. During the $j^{th}$ round, the active set $\mathcal{A}_j(t)$ contains all points that have not been eliminated from being the nearest neighbor of $x_j$ at time $t$. In what follows, we add a superscript $\triangle$ to denote a bound obtained via the triangle inequality, whose precise definitions are given in Lemma 3.1. We define $x_j$'s active set at time $t$ as

$$\mathcal{A}_j(t) := \{a \neq j : \max\{L_{a,j}(t), L_{a,j}^{\triangle}(t)\} < \min_k \min\{U_{j,k}(t), U_{j,k}^{\triangle}(t)\}\}. \tag{5}$$

Assuming $L_{a,j}^{\triangle}(t)$ and $U_{j,k}^{\triangle}(t)$ are valid lower and upper bounds on $d_{a,j}, d_{j,k}$ respectively, (5) states that point $x_a$ is active if its lower bound is less than the minimum upper bound for $d_{j,k}$ over all choices of $x_k \neq x_j$. Next, for any $(j,k)$ we construct triangle bounds $L^{\triangle}, U^{\triangle}$ on the distance $d_{j,k}$. Intuitively, for some reals $g, g', h, h'$, if $d_{i,j} \in [g, g']$ and $d_{i,k} \in [h, h']$ then $d_{j,k} \leq g' + h'$, and

$$d_{j,k} \geq |d_{i,j} - d_{i,k}| = \max\{d_{i,j}, d_{i,k}\} - \min\{d_{i,j}, d_{i,k}\} \geq (\max\{g, h\} - \min\{g', h'\})_+ \tag{6}$$

where $(s)_+ := \max\{s, 0\}$. The lower bound can be seen as true by Fig. 7 in the Appendix. Lemma 3.1 uses these ideas to form upper and lower bounds on distances by the triangle inequality. Note that this definition is inherently recursive as it may rely on past triangle inequality bounds to achieve the tightest possible result. We denote a triangle inequality upper and lower bounds on $d_{j,k}$ due to a point $i$ at time $t$ as $U_{j,k}^{\triangle_i}$ and $L_{j,k}^{\triangle_i}$ respectively.

**Lemma 3.1.** *For all $k \neq 1$, $U_{1,k}^{\triangle_1}(t) = U_{1,k}^{\triangle}(t) = U_{1,k}(t)$. For any $i < j$ define*

$$U_{j,k}^{\triangle_i}(t) := \min_{\max\{i_1, i_2\} < i}(\min\{U_{i,j}(t), U_{i,j}^{\triangle_{i_1}}(t)\} + \min\{U_{i,k}(t), U_{i,k}^{\triangle_{i_2}}(t)\}). \tag{7}$$

*For all $k \neq 1$, $L_{1,k}^{\triangle_1}(t) = L_{1,k}^{\triangle}(t) = L_{1,k}(t)$. For any $i < j$ define*

$$L_{j,k}^{\triangle_i}(t) := \max_{\max\{i_1, i_2, i_3, i_4\} < i}\Big( \max\{L_{i,j}(t), L_{i,j}^{\triangle_{i_1}}(t), L_{i,k}(t), L_{i,k}^{\triangle_{i_2}}(t)\} $$
$$- \min\{U_{i,j}(t), U_{i,j}^{\triangle_{i_3}}(t), U_{i,k}(t), U_{i,k}^{\triangle_{i_4}}(t)\}\Big)_+, \tag{8}$$

*where $(s)_+ := \max\{s, 0\}$. If all the bounds obtained by SETri in rounds $i < j$ are correct then*

$$d_{j,k} \in \big[L_{j,k}^{\triangle}(t), U_{j,k}^{\triangle}(t)\big], \quad \text{where} \quad L_{j,k}^{\triangle}(t) := \max_{i<j} L_{j,k}^{\triangle_i}(t) \quad \text{and} \quad U_{j,k}^{\triangle}(t) := \min_{i<j} U_{j,k}^{\triangle_i}(t).$$

The proof is in Appendix B.1. ANNTri has access to two sources of bounds on distances: concentration bounds and triangle inequality bounds, and as can be seen in Lemma 3.1, the former affects the latter. Furthermore, triangle bounds are computed from other triangle bounds, leading to the recursive definitions of $L_{j,k}^{\triangle_i}$ and $U_{j,k}^{\triangle_i}$. Because of these facts, triangle bounds are dependent on the order in which ANNTri finds each nearest neighbor. These bounds can be computed using dynamic programming and brute force search over all possible $i_1, i_2, i_3, i_4$ is not necessary. We note that the above recursion is similar to the Floyd-Warshall algorithm for finding shortest paths between all pairs of vertices in a weighted graph [11, 7]. The results in [27] show that the triangle bounds obtained in this manner have the minimum $L_1$ norm between the upper and lower bound matrices.

**Extension to $k$-Nearest Neighbor Graphs:** All algorithms and theory in this paper can be extended to the case of $k$-nearest neighbor graphs where one wishes to draw a directed edge from each point to all of its $k$-nearest neighbors. To modify ANNTri, one can change the subroutine SETri to be a variant of the KL-Racing algorithm by [21], for instance. Racing style algorithms are the natural extension of Successive-Elimination style algorithms to the top-$k$ bandit setting. To achieve best complexity, in this case, one would again want to sample the distances with the only minimum number of calls to the oracle first, as in Line 3 of SETri. To make a statement similar to Theorem 4.4, i.e. to bound the complexity of learning $k$-NN graphs, it is necessary to alter the definition of the events $A_{j,k}$ so as to certify that a point has been eliminated from being a $k$-nearest neighbor as opposed to a 1-nearest neighbor in the current form. The suboptimality gaps $\Delta_{j,k}$ (and therefore $H_{j,k}$) would be defined differently for the $k$-nearest neighbor case leading to a different bound. A similar statement as Theorem 4.6 can likewise be achieved as long as $k < \log(n)$ and one should expect a complexity of $\mathcal{O}\left(kn\log(n)\overline{\Delta^{-2}}\right)$ for an appropriately defined $\overline{\Delta^{-2}}$.

# 4 Analysis

All omitted proofs of this section can be found in the Appendix Section B.

**Theorem 4.1.** *ANNTri finds the nearest neighbor for each point in $\mathcal{X}$ with probability $1 - \delta$.*

## 4.1 A simplified algorithm

The following Lemma indicates which points must be eliminated initially in the $j^{th}$ round.

**Lemma 4.2.** *If $\exists i : 2U_{i,j} < L_{i,k}$, then $x_k \notin \mathcal{A}_j(0)$ for ANNTri.*

*Proof.* $2U_{i,j} < L_{i,k} \iff U_{i,j} < L_{i,k} - U_{i,j} \le L_{j,k}^{\triangle_i}$ $\qquad\qquad\qquad\qquad\square$

Next, we define ANNEasy, a simplified version of ANNTri that is more amenable to analysis. Here, we say that $x_k$ is eliminated in the $j^{th}$ round of ANNEasy if i) $k<j$ and $\exists i : U_{i,j} < L_{j,k}$ (symmetry from past samples) or ii) $\exists i : 2U_{i,j} < L_{i,k}$ (Lemma 4.2). Therefore, $x_j$'s active set for ANNEasy is

$$\mathcal{A}_j = \{a \neq j : L_{a,k} \le 2U_{j,k} \; \forall k \;\; \text{and} \;\; L_{a,j} < \min_k U_{j,k}\}. \tag{9}$$

To define ANNEasy in code, we remove lines 3-6 of ANNTri (Algorithm 1), and call a subroutine SEEasy in place of SETri. SEEasy matches SETri (Algorithm 2) except that lines 1 and 8 are replaced with (9) instead. We provide full pseudocode of both ANNEasy and SEEasy in the Appendix A.1.1. Though ANNEasy is a simplification for analysis, we note that it empirically captures much of the same behavior of ANNTri. In the Appendix A.1.2 we provide an empirical comparison of the two.

## 4.2 Complexity of ANNEasy

We now turn our attention to account for the effect of the triangle inequality in ANNEasy.

**Lemma 4.3.** *For any $x_k \in \mathcal{X}$ if the following conditions hold for some $i < j$, then $x_k \notin \mathcal{A}_j(0)$.*

$$6C_{\delta/n}(1) \le d_{i,k} - 2d_{i,j} \quad \text{and} \quad \{j,k\} \cap (\cup_{m<i}\{\ell : 2d_{m,i} < d_{m,\ell}\}) = \emptyset. \tag{10}$$

The first condition characterizes which $x_k$'s must satisfy the condition in Lemma 4.2 for the $j^{th}$ round. The second guarantees that $x_k$ was sampled in the $i^{th}$ round, a necessary condition for forming triangle bounds using $x_i$.

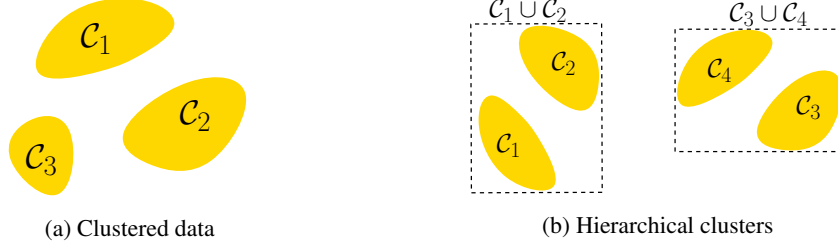

<div align="center">

(a) Clustered data            (b) Hierarchical clusters

Figure 1: Example datasets where triangle inequalities lead to provable gains.

</div>

**Theorem 4.4.** *Conditioned on the event that all confidence bounds are valid at all times, `ANNEasy` learns the nearest neighbor graph of $\mathcal{X}$ in the following number of calls to the distance oracle:*

$$\mathcal{O}\left( \sum_{j=1}^{n} \sum_{k>j} \mathbb{1}_{[A_{j,k}]} H_{j,k} + \sum_{k<j} \mathbb{1}_{[A_{j,k}]} (H_{j,k} - \mathbb{1}_{[A_{k,j}]} H_{k,j})_+ \right). \tag{11}$$

*In the above expression $H_{j,k} := \frac{\log(n^2/(\delta\Delta_{j,k}))}{\Delta_{j,k}^2}$ and $\mathbb{1}_{[A_{j,k}]} := 1$, if $x_k$ does not satisfy the triangle inequality elimination conditions of* (10) $\forall i < j$, *and* 0 *otherwise.*

The expression in (11) can be understood as a sum over the complexity of each of the $n$ rounds, as specified by the outer sum. The complexity of each individual round is a sum of two terms. Consider the $j^{th}$ round. The first term bounds the number of calls to $\mathsf{Q}(j,k)$ for all $k > j$. In general $H_{j,k}$ calls are necessary, unless a triangle inequality bound allows for elimination of $k$ without sampling, as given by $\mathbb{1}_{[A_{j,k}]}$. The second term bounds the number of calls to $\mathsf{Q}(j,k)$ for all $k < j$. It has the same form as the first term, except we must now use past samples we may already have via symmetry of distances (provided the triangle inequality did not prevent us from querying $\mathsf{Q}(k,j)$ in the previous round). The $(\cdot)_+$ operation prevents negative terms, since it may be the case that no additional samples are necessary, even if we don't use the triangle inequality for elimination.

In Theorem B.6, in the Appendix, we state the sample complexity when triangle inequality bounds are ignored by `ANNTri`, and this upper bounds (11). Whether a point can be eliminated by the triangle inequality depends both on the underlying distances and the order in which `ANNTri` finds each nearest neighbor (*c.f.* Lemma 4.3). In general, this dependence on the order is necessary to ensure that past samples exist and may be used to form upper and lower bounds. Furthermore, it is worth noting that even without noise the triangle inequality may not always help. A simple example is any arrangement of points such that $0 < r \leq d_{j,k} < 2r \;\forall j, k$. To see this, consider triangle bounds on any distance $d_{j,k}$ due to any $x_i, x_{i'} \in \mathcal{X}\backslash\{x_j, x_k\}$. Then $|d_{i,j} - d_{i,k}| \leq r < 2r \leq d_{i',j} + d_{i',k} \;\forall i, i'$ so $L_{i,j}^{\triangle} < U_{j,k}^{\triangle} \;\forall i, j, k$. Thus no triangle upper bounds separate from triangle lower bounds so no elimination via the triangle inequality occurs. In such cases, it is necessary to sample all $\mathcal{O}(n^2)$ distances. However, in more favorable settings where data may be split into clusters, the sample complexity can be much lower by using triangle inequality.

The order in which $\{x_{i^*}\}$ are found follows their subscript index, which is randomly chosen and fixed before starting the algorithm. As described above, different orders in which $\{x_i\}$ are processed can affect the query complexity of our algorithm. The best order that minimizes the total number of queries made in general depends on the true distance values. Even if the oracle is noiseless, there are datasets where the pair $(i, j)$ with the smallest $d_{i,j}$ must be queried within the first $n$ queries in order to identify the NN-graph using the minimum number of queries. Since this requirement cannot be ensured by any algorithm that only has access to information via a distance oracle, it is not possible to achieve the minimum number of queries in such examples.

## 4.3 Adaptive gains via the triangle inequality

We highlight two settings where `ANNTri` provably achieves sample complexity better than $\mathcal{O}(n^2)$ independent of the order of the rounds. Consider a dataset containing $c$ clusters of $n/c$ points each as in Fig. 1a. Denote the $m^{th}$ cluster as $\mathcal{C}_m$ and suppose the distances between the points are such that

$$\{x_k : d_{i,k} < 6C_{\delta/n}(1) + 2d_{i,j}\} \subseteq \mathcal{C}_m \;\forall i, j \in \mathcal{C}_m. \tag{12}$$

The above condition is ensured if the distance between any two points belonging to different clusters is at least a $(\delta, n)$-dependent constant plus twice the diameter of any cluster.

**Theorem 4.5.** *Consider a dataset of $\sqrt{n}$ clusters which satisfy the condition in (12). Then* `ANNEasy` *learns the correct nearest neighbor graph of $\mathcal{X}$ with probability at least $1 - \delta$ in*

$$\mathcal{O}\left(n^{3/2}\overline{\Delta^{-2}}\right) \tag{13}$$

*queries where $\overline{\Delta^{-2}} := \frac{1}{n^{3/2}} \sum_{i=1}^{\sqrt{n}} \sum_{j,k \in \mathcal{C}_i} \log(n^2/(\delta\Delta_{j,k}))\Delta_{j,k}^{-2}$ is the average number of samples distances between points in the same cluster.*

By contrast, random sampling requires $\mathcal{O}(n^2\Delta_{\min}^{-2})$ where $\Delta_{\min}^{-2} := \min_{j,k} \log(n^2/(\delta\Delta_{j,k}))\Delta_{j,k}^{-2} \geq \overline{\Delta^{-2}}$. In fact, the value in (11) can be even lower if unions of clusters also satisfy (12). In this case, the triangle inequality can be used to separate *groups* of clusters. For example, in Fig. 1b, if $\mathcal{C}_1 \cup \mathcal{C}_2$ and $\mathcal{C}_3 \cup \mathcal{C}_4$ satisfy (12) along with $\mathcal{C}_1, \cdots, \mathcal{C}_4$, then the triangle inequality can separate $\mathcal{C}_1 \cup \mathcal{C}_2$ and $\mathcal{C}_3 \cup \mathcal{C}_4$. This process can be generalized to consider a dataset that can be split recursively into subclusters following a binary tree of $k$ levels. At each level, the clusters are assumed to satisfy (12). We refer to such a dataset as *hierarchical in* (12).

**Theorem 4.6.** *Consider a dataset $\mathcal{X} = \cup_{i=1}^{n/\nu}\mathcal{C}_i$ of $n/\nu$ clusters of size $\nu = \mathcal{O}(\log(n))$ that is hierarchical in (12). Then* `ANNEasy` *learns the correct nearest neighbor graph of $\mathcal{X}$ with probability at least $1 - \delta$ in*

$$\mathcal{O}\left(n\log(n)\overline{\Delta^{-2}}\right) \tag{14}$$

*queries where $\overline{\Delta^{-2}} := \frac{1}{n\nu} \sum_{i=1}^{n/\nu} \sum_{j,k \in \mathcal{C}_i} \log(n^2/(\delta\Delta_{j,k}))\Delta_{j,k}^{-2}$ is the average number of samples distances between points in the same cluster.*

Expression (14) matches known lower bounds of $\mathcal{O}(n\log(n))$ on the sample complexity for learning the nearest neighbor graph from noiseless samples [30], the additional penalty of $\overline{\Delta^{-2}}$ is due to the effect of noise in the samples. An easy way to see the lower bound is to consider the fact that there are $\mathcal{O}(n^{n-1})$ unique nearest neighbor graphs so any algorithm will require $\mathcal{O}(\log(n^{n-1})) = \mathcal{O}(n\log(n))$ bits of information to identify the correct one. In Appendix C, we state the sample complexity in the average case, as opposed to the high probability statements above. The analog of the cluster condition (12) there does not involve constants and is solely in terms of pairwise distances (*c.f.* (33)).

## 5 Experiments

Here we evaluate the performance of `ANNTri` on simulated and real data. To construct the tightest possible confidence bounds for `SETri`, we use the law of the iterated logarithm as in [18] with parameters $\epsilon = 0.7$ and $\delta = 0.1$. Our analysis bounds the number of queries made to the oracle. We visualize the performance by tracking the empirical *error rate* with the number of queries made per point. For a given point $x_i$, we say that a method makes an error at the $t^{th}$ sample if it fails to return $x_{i*}$ as the nearest neighbor, that is, $x_{i*} \neq \arg\min_j \hat{d}[i,j]$. Throughout, we will compare `ANNTri` against random sampling. Additionally, to highlight the effect of the triangle inequality, we will compare our method against the same active procedure, but ignoring triangle inequality bounds (referred to as `ANN` in plots). All baselines may reuse samples via symmetry as well. We plot all curves with $95\%$ confidence regions shaded.

### 5.1 Simulated Experiments

We test the effectiveness of our method, we generate an embedding of 10 clusters of 10 points spread around a circle such that each cluster is separated by at least $10\%$ of its diameter in $\mathbb{R}^2$ as in shown in Fig. 2a. We consider Gaussian noise with $\sigma = 0.1$. In Fig. 2b, we present average error rates of `ANNTri`, `ANN`, and `Random` plotted on a log scale. `ANNTri` quickly learns $x_{i*}$ and has lower error with 0 samples due to initial elimination by the triangle inequality. The error curves are averaged over 4000 repetitions. All rounds were capped at $10^5$ samples for efficiency.

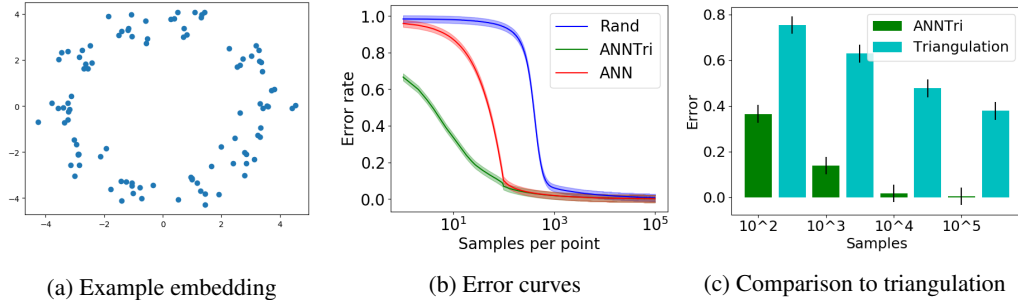

|   |   |   |
|---|---|---|
| (a) Example embedding | (b) Error curves | (c) Comparison to triangulation |

Figure 2: Comparison of `ANNTri` to `ANN` and `Random` for 10 clusters of 10 points separated by $10\%$ of their diameter with $\sigma = 0.1$. `ANNTri` identifies clusters of nearby points more easily.

### 5.1.1 Comparison to triangulation

An alternative way a practitioner may use to obtain the nearest neighbor graph might be to estimate distances with respect to a few anchor points and then triangulate to learn the rest. [9] provide a comprehensive example, and we summarize in Appendix A.2 for completeness. The triangulation method is naïve for two reasons. First, it requires *much* stronger modeling assumptions than `ANNTri`— namely that the metric is Euclidean and the points are in a low-dimensional of known dimension. Forcing Euclidean structure can lead to unpredictable errors if the underlying metric might not be Euclidean, such as in data from human judgments. Second, this procedure may be more noise sensitive because it estimates squared distances. In the example in Section A.2, this leads to the additive noise being sub-exponential rather than subGaussian. In Fig. 2c, we show that even in a favorable setting where distances are truly sampled from a low-dimensional Euclidean embedding and pairwise distances between anchors are known exactly, triangulation still performs poorly compared to `ANNTri`. We consider the same 2-dimensional embedding of points as in Fig. 2a for a noise variance of $\sigma = 1$ and compare the `ANNTri` and triangulation for different numbers of samples.

## 5.2 Human judgment experiments

### 5.2.1 Setup

Here we consider the problem of learning from human judgments. For this experiment, we used a set $\mathcal{X}$ of 85 images of shoes drawn from the UT Zappos50k dataset [32, 33] and seek to learn which shoes are most visually similar. To do this, we consider queries of the form "between $i$, $j$, and $k$, which two are most similar?". We show example queries in Figs. 5a and 5b in the Appendix. Each query maps to a pair of triplet judgments of the form "is $j$ or $k$ more similar to $i$?". For instance, if $i$ and $j$ are chosen, then we may imply the judgments "$i$ is more similar to $j$ than to $k$" and "$j$ is more similar to $i$ than to $k$". We therefore construct these queries from a set of triplets collected from participants on Mechanical Turk by [15]. The set contains multiple samples of all $85\binom{84}{2}$ unique triples so that the probability of any triplet response can be estimated. We expect that $i^*$ is most commonly selected as being more similar to $i$ than any third point $k$. We take distance to correspond to the fraction of times that two images $i$, $j$ are judged as being more similar to each other than a different pair in a triplet query $(i, j, k)$. Let $E_{i,k}^{j}$ be the event that the pair $i, k$ are chosen as most similar amongst $i$, $j$, and $k$. Accordingly, we define the 'distance' between images $i$ and $j$ as

$$d_{i,j} := \mathbb{E}_{k \sim \text{Unif}(\mathcal{X} \setminus \{i,j\})} \mathbb{E}[\mathbb{1}_{E_{i,k}^{j}} | k]$$

where $k$ is drawn uniformly from the remaining 83 images in $\mathcal{X} \setminus \{i, j\}$. For a fixed value of $k$,

$$\mathbb{E}[\mathbb{1}_{E_{i,k}^{j}} | k] = \mathbb{P}(E_{i,k}^{j} | k) = \mathbb{P}(\text{"}i \text{ more similar to } j \text{ than to } k\text{"})\mathbb{P}(\text{"}j \text{ more similar to } i \text{ than to } k\text{"}).$$

where the probabilities are the empirical probabilities of the associated triplets in the dataset. This distance is a quasi-metric on our dataset as it does not always satisfy the triangle inequality; but satisfies it with a multiplicative constant: $d_{i,j} \leq 1.47(d_{i,k} + d_{j,k}) \ \forall i, j, k$. Relaxing metrics to quasi-metrics has a rich history in the classical nearest neighbors literature [16, 29, 14], and `ANNTri` can be trivially modified to handle quasi-metrics. However, we empirically note that $< 1\%$ of the distances violate the ordinary triangle inequality here so we ignore this point in our evaluation.

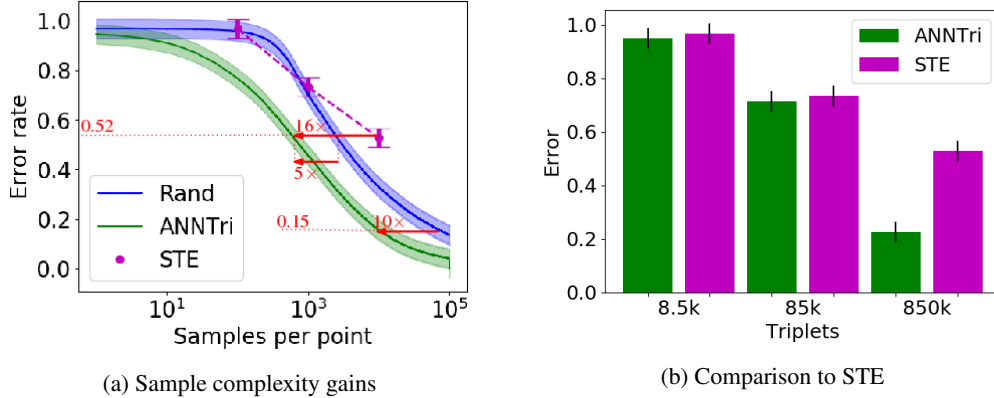

(a) Sample complexity gains          (b) Comparison to STE

Figure 3: Performance of `ANNTri` on the Zappos dataset. `ANNTri` achieves superior performance over STE in identifying nearest neighbors and has $5 - 10$x gains in sample efficiency over random.

### 5.2.2 Results

When `ANNTri` or any baseline queries $Q(i, j)$ from the oracle, we randomly sample a third point $k \in \mathcal{X} \backslash \{i, j\}$ and flip a coin with probability $\mathbb{P}(E_{i,k}^j)$. The resulting sample is an unbiased estimate of the distance between $i$ and $j$. In Fig. 3a, we compare the error rate averaged over 1000 trials of `ANNTri` compared to `Random` and STE. We also plot associated gains in sample complexity by `ANNTri`. In particular, we see gains of $5 - 10$x over random sampling, and gains up to 16x relative to ordinal embedding. `ANNTri` also shows 2x gains over `ANN` in sample complexity (see Fig. 6 in Appendix).

Additionally, a standard way of learning from triplet data is to perform ordinal embedding. With a learned embedding, the nearest neighbor graph may easily be computed. In Fig. 3b, we compare `ANNTri` against the state of the art STE algorithm [31] for estimating Euclidean embeddings from triplets, and select the embedding dimension of $d = 16$ via cross validation. To normalize the number of samples, we first perform `ANNTri` with a given max budget of samples and record the total number needed. Then we select a random set of triplets of the same size and learn an embedding in $\mathbb{R}^{16}$ via STE. We compare both methods on the fraction of nearest neighbors predicted correctly. On the $x$ axis, we show the total number of triplets given to each method. For small dataset sizes, there is little difference, however, for larger dataset sizes, `ANNTri` significantly outperforms STE. Given that `ANNTri` is active, it is reasonable to wonder if STE would perform better with an actively sampled dataset, such as [28]. Many of these methods are computationally intensive and lack empirical support [20], but we can embed using the full set of triplets to mitigate the effect of the subsampling procedure. Doing so, STE achieves $52\%$ error, within the confidence bounds of the largest subsample shown in Fig. 3b. In particular, more data and more carefully selected datasets, may not correct for the bias induced by forcing Euclidean structure.

## 6 Conclusion

In this paper we solve the nearest neighbor graph problem by adaptively querying distances. Our method makes no assumptions beyond standard metric properties and is empirically shown to achieve sample complexity gains over passive sampling. In the case of clustered data, we show provable gains and achieve optimal rates in favorable settings. One interesting avenue for future work would be to specialize to the case of hyperbolic embeddings which naturally encode trees [8] and may be a more flexible way to describe hierarchical clusters as in Theorem 4.6. Implementations of `ANNTri`, `ANN`, and `RANDOM` can be found alongside a demo and summary slides at https://github.com/blakemas/nngraph.

## Acknowledgments

The authors wish to thank Lalit Jain for many helpful discussions over the course of this work for which the paper is better and the reviewers for their helpful suggestions. This work was partially supported by AFOSR/AFRL grants FA8750-17-2-0262 and FA9550-18-1-0166.

## Footnotes

*Authors contributed equally to this paper and are listed alphabetically.

[1] We could also proceed in a non-iterative manner, by adaptively choosing which among $\binom{n}{2}$ pairs to query next. However this has worse empirical performance and same theoretical guarantees as the in-order approach.

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
