[Supplementary Material]

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

      matrix where each entry is $-\infty$, NN as a length $n$ array
  2: **for** $j = 1$ **to** $n$ **do**
  3:    NN$[j]$ = `SEEasy`$(j, \hat{d}, U, L, T, \xi = \delta/n)$
  4: **return** The nearest neighbor graph adjacency list NN

---

---

**Algorithm 4** `SEEasy`

---

**Require:** index $j$, callable oracle $\mathsf{Q}(\cdot, \cdot)$ (1), 4 $n \times n$ matrices: $\hat{d}, U, L, T$, confidence $\xi$
  1: Initialize the active set $\mathcal{A}_j \leftarrow \{a \neq j : L[a,k] \leq 2U[j,k] \,\forall k \text{ and } L[a,j] < \min_k U[j,k]\}$
  2: **while** $|\mathcal{A}_j| > 1$ **do**
  3:    **for all** $i \in \mathcal{A}_j$ such that $T[i,j] = \min_{k \in \mathcal{A}_j} T[i,k]$ **do** {only query points with fewest samples}
  4:       Update $\hat{d}[i,j]$, $\hat{d}[j,i] \leftarrow (\hat{d}[i,j] \cdot T[i,j] + \mathsf{Q}(i,j))/(T[i,j]+1)$
  5:       Update $T[i,j]$, $T[j,i] \leftarrow T[i,j] + 1$
  6:       Update $U[i,j]$, $U[j,i] \leftarrow \hat{d}[i,j] + C_\xi(T[i,j])$
  7:       Update $L[i,j]$, $L[j,i] \leftarrow \hat{d}[i,j] - C_\xi(T[i,j])$
  8:    Update $\mathcal{A}_j \leftarrow \{a \neq j : L[a,k] \leq 2U[j,k] \,\forall k \text{ and } L[a,j] < \min_k U[a,k]\}$
  9: **return** The index $i$ for which $x_i \in \mathcal{A}_j$

---

# Appendix

# A   Additional experimental results and details

## A.1   Differences between `ANNTri` and `ANNEasy`

### A.1.1   Pseudocode for `ANNEasy` and `SEEasy`

We begin by providing pseudocode for both `ANNEasy` and `SEEasy` as described in Section 4.1 in Algorithms 3 and 4.

### A.1.2   Empirical differences in performance for `ANNTri` and `ANNEasy`

In Figure 4 we compare the empirical performance of `ANNTri` and `ANNEasy`. We compare their performance in the same setting as Figure 2a with 10 clusters of 10 points separated by their at least 10% of their diameter. The curves are averaged over 4000 independent trials and plotted with 95% confidence regions. As is indicated in the plot, `ANNEasy` has similar behavior as `ANNTri`, but achieves slightly worse performance.

## A.2   Triangulation

In this section, we provide a brief review of triangulation to estimate Euclidean embeddings, similar to the presentation in [9]. The method is summarized as follows. Let $\mathcal{X}$ be a set of $n$ points in Euclidean $d$ space and $\boldsymbol{D}$ be the associated Euclidean distance matrix where each entry is the square of the associated Euclidean distance. Let $A$ be a set of anchor points. Without loss of generality, we take $A := \{x_1, \cdots, x_{d+2}\}$. The $+2$ is to correct for the fact that Euclidean distance matrices have rank $d + 2$. Let $\boldsymbol{A} := \boldsymbol{D}[1 : d+2, 1 : d+2]$ and $\boldsymbol{L} := \boldsymbol{D}[1 : d+2, 1 : n]$. Then it can easily be verified that $\boldsymbol{D} = \boldsymbol{L}\boldsymbol{A}^{-1}\boldsymbol{L}^T$. To learn the entries in $\boldsymbol{L}$ (as well as $\boldsymbol{A}$), sample the distance from each of the $n$ points to the $d + 2$ anchors as many times as there is budget for and square the results. The empirical mean is a plugin estimator of the associated entry in $\boldsymbol{L}$ and $\boldsymbol{A}$, and we take $\widehat{\boldsymbol{L}}$ and $\widehat{\boldsymbol{A}}$ to be their unbiased estimates. Therefore $\widehat{\boldsymbol{D}} := \widehat{\boldsymbol{L}}\widehat{\boldsymbol{A}}^{-1}\widehat{\boldsymbol{L}}^T$ is an unbiased estimate of $\boldsymbol{D}$. With $\widehat{\boldsymbol{D}}$, the nearest neighbor graph can easily be computed.

Figure 4: Comparison of error in identifying $x_{i*}$ `ANNTri` and the `ANNEasy` for $10$ clusters of $10$ points separated by $10\%$ of their diameter with $\sigma = 0.1$.

Click on the two most similar shoes

Click on the two most similar shoes

(a) An easy query

(b) A harder query

Figure 5: Two example zappos queries.

## A.3 Additional experimental results for Zappos dataset

In Fig. 5 we show two example queries of the form "which pair are most similar of these three?". Some queries are more straightforward whereas some are more subjective.

Additionally, in Fig. 6, we show the performance of `ANNTri`, `ANN`, and `Random` in identifying nearest neighbors from the Zappos data. In this setting, there is less of an advantage to using the triangle inequality due to the highly noisy and subjective nature of human judgments. Despite this, we still see a slight advantage to `ANNTri` over `ANN`. In particular, for moderate accuracy, there is a gain sample complexity of around 2x.

## B   Proofs and technical lemmas

### B.1   Proof of Lemma 3.1

By symmetry for all $i < j$, we have existing samples of $\mathsf{Q}(i,j)$ and $\mathsf{Q}(i,k)$ and we use bounds based on these samples as well as past triangle inequality upper bounds on $d_{i,j}$ and $d_{i,k}$ due to $i_1 < i$ and $i_2 < i$ respectively. The upper bound is derived as follows:

$$d_{j,k} \leq d_{i,j} + d_{i,k} \leq \min\{U_{i,j}(t), U_{i,j}^{\triangle_{i_1}}(t)\} + \min\{U_{i,k}(t), U_{i,k}^{\triangle_{i_2}}(t)\} =: U_{j,k}^{\triangle_i}$$

Since we may form bounds based on all $i < j$ for which we have both samples of $\mathsf{Q}(i,j)$ and $\mathsf{Q}(i,k)$, we may optimize over $i$ to get the tightest possible triangle inequality bounds on $d_{j,k}$.

Lower bounds are derived similarly. Again, intuitively, we may use past samples of both $\mathsf{Q}(i,j)$ and $\mathsf{Q}(i,k)$ and associated bounds to derive a lower bound on $d_{j,k}$. The form is slightly more complicated here since we have to worry about both upper and lower bounds on $d_{i,j}$ and $d_{i,k}$. These bounds may

Figure 6: Error rates for nearest neighbor identification on Zappos Data

either be from concentration bounds based on past samples directly or past triangle inequality upper and lower bounds on these distances due to points $i_1 - i_4 < i$.

$$
\begin{aligned}
d_{j,k} \geq & |d_{i,j} - d_{i,k}| \\
= & \max\{d_{i,j}, d_{i,k}\} - \min\{d_{i,j}, d_{i,k}\} \\
\geq & (\max\{\max\{L_{i,j}(t), L_{i,j}^{\triangle_{i_1}}(t)\},\ \max\{L_{i,k}(t), L_{i,k}^{\triangle_{i_2}}(t)\}\} \\
& - \min\{\min\{U_{i,j}(t), U_{i,j}^{\triangle_{i_3}}(t)\},\ \min\{U_{i,k}(t), U_{i,k}^{\triangle_{i_4}}(t)\}\})_+ \\
= & (\max\{L_{i,j}(t), L_{i,j}^{\triangle_{i_1}}(t), L_{i,k}(t), L_{i,k}^{\triangle_{i_2}}(t)\} \\
& - \min\{U_{i,j}(t), U_{i,j}^{\triangle_{i_3}}(t), U_{i,k}(t), U_{i,k}^{\triangle_{i_4}}(t)\})_+
\end{aligned}
$$

where $(s)_+ := \max\{s, 0\}$ and $i_1, i_2, i_3, i_4 < i$, (not necessarily unique) are chosen to optimize the bound. Similar to the upper bound, this holds with respect to any $i < j$ and we optimize over $i$. To ease presentation, let $\text{UB}'[i,j] := \min\{U_{i,j}, \min_{l<i} U_{i,j}^{\triangle_l}\}$ and $\text{LB}'[i,j] := \max\{L_{i,j}, \max_{l<i} L_{i,j}^{\triangle_l}\}$ be the tightest upper and lower bounds for $d_{i,j}$. For the lower bound, note that if the argument of $(\cdot)_+$ is negative, then any

$$
\begin{aligned}
s \in & [\max\{\text{LB}'[i,j], \text{LB}'[i,k]\}, \min\{\text{UB}'[i,j], \text{UB}'[i,k]\}] \\
= & [\text{LB}'[i,j], \text{UB}'[i,j]] \cap [\text{LB}'[i,k], \text{UB}'[i,k]] \neq \emptyset
\end{aligned}
$$

can be the value of both $d_{i,j}$ and $d_{j,k}$ as it lies in both their confidence intervals. Then points $x_j, x_k$ can possibly be at the same location in the metric space, in which case $d_{j,k} = 0$. On the other hand if the RHS is positive, then $x_j$ and $x_k$ cannot be at the same location as $d_{i,j} \neq d_{i,k}$. In fact, the smallest possible value for $d_{j,k}$ occurs if $x_i, x_j, x_k$ are collinear. This can be seen to be true from Figure 7. We finish with a quick lemma noting what can and cannot be eliminated via the triangle inequality.

**Lemma B.1.** *Conditioned on the good event that all bounds are correct at all times, the triangle inequality cannot be used to to separate the two closest points to any given third point.*

*Proof.* Consider finding $x_{i^*}$. Let $d_{i,i^*} \leq d_{i,j} \leq d_{i,k} \forall k \neq i^*, j$. By the triangle inequality, $d_{i,i^*} \leq d_{i,j} + d_{j,i^*}$ Clearly, the RHS is no smaller than $d_{i,j}$. Since we are conditioning on all bounds being correct at all times, no upper bound on $d_{i,i^*}$ from the triangle inequality can ever be smaller that $d_{i,j}$. Rearranging the inequality, we see that $d_{i,i^*} - d_{j,i^*} \leq d_{i,j}$. The LHS is no larger than $d_{i,i^*}$, and $d_{i,i^*}$

Figure 7: Pictorial justification for the lower bound in (6). True positions of points $i, j, k$ are shown along with the upper and lower bounds for $d_{i,j}, d_{i,k}$ that are known to the algorithm. If the angle $\theta$ between $\vec{ij}$ and $\vec{ik}$ is known, the blue segment shows the lowest possible value for $d_{j,k}$ based on the bounds. The orange segment is the value in the RHS of (6). Without any information about $\theta$, the three points could be collinear, in which case $d_{j,k}$ could equal the length of the orange segment.

is the only distance wrt $x_i$ that is smaller than $d_{i,j}$ by assumption. Therefore, no lower bound on $d_{i,j}$ due to the triangle inequality is greater than $d_{i,i^*} < d_{i,j}$. $\qquad \square$

### B.1.1 Helper Lemmas

**Lemma B.2.** *Let $t \in \mathbb{N}$ index the rounds of the procedure SETri in finding $x_{i^*}$. Suppose all confidence intervals are valid, i.e., (3) is true. Then $\forall j \neq i$ and all $t$,*

$$L_{i,j}(t) \geq d_{i,j} - 2C_{\delta/n}(T_{i,j}(t)) \quad and \quad U_{i,j}(t) \leq d_{i,j} + 2C_{\delta/n}(T_{i,j}(t)). \tag{15}$$

*Proof.* If the good event (3) is true then for any pair $(i, j)$ and time $t$ we have

$$\hat{d}_{i,j}(t) < d_{i,j} + C_{\delta/n}(T_{i,j}(t)) \implies U_{i,j}(t) := \hat{d}_{i,j}(t) + C_{\delta/n}(T_{i,j}(t)) \leq d_{i,j} + 2C_{\delta/n}(T_{i,j}(t)).$$

A similar calculation can be done for $L_{i,j}(t)$ as well. $\qquad \square$

**Lemma B.3.** *Let $j > i$, and let $t_j$ be the time when $x_j$ is last sampled in the $i^{th}$ round and equivalently for $t_k$. Assume without loss of generality that $d_{i,j} < d_{i,k}$. If $d_{i,j}$ and $d_{i,k}$ are such that*

$$4C_{\delta/n}(T_{i,j}(t_j)) + 2C_{\delta/n}(T_{i,k}(t_k)) \leq d_{i,k} - 2d_{i,j} \tag{16}$$

*then SETri can eliminate $d_{j,k}$ without sampling it, i.e., $x_k \notin \mathcal{A}_j(0)$.*

*Proof.* Focusing on the number of $\mathsf{Q}(i,j)$ queries, we have that

$$2U_{i,j}(t_j) = 2(\hat{d}_{i,j}(t_j) + C_{\delta/n}(T_{i,j}(t_j))) \leq 2(d_{i,j} + 2C_{\delta/n}(T_{i,j}(t_j))), \tag{17}$$

the inequality in (17) is due to Lemma B.2, and using the number of $\mathsf{Q}(i,k)$ queries,

$$L_{i,k}(t_k) \geq \hat{d}_{i,k}(t_k) - C_{\delta/n}(T_{i,k}(t_k)) \geq d_{i,k} - 2C_{\delta/n}(T_{i,k}(t_k)). \tag{18}$$

The first inequality in (18) is because if $k < j$ then there may have been more $\mathsf{Q}(k,i)$ queries beyond the $t_k$ number of $\mathsf{Q}(i,k)$ queries made while finding $x_{i^*}$. Rearranging the equation in the Lemma statement,

$$2d_{i,j} + 4C_{\delta/n}(T_{i,j}(t_j)) \leq d_{i,k} - 2C_{\delta/n}(T_{i,k}(t_k)),$$

which implies that $2U_{i,j} \leq L_{i,k}$ from (17), (18). Hence from Lemma 4.2 $x_k \notin \mathcal{A}_j(0)$. $\qquad \square$

**Lemma B.4.** *There exists a dataset $\mathcal{P}$ containing $2\nu$ points such that for all $x_p \in \mathcal{P}$ and $\alpha > 0$ the set of suboptimality gaps $\Delta_{p,p'}$ is*

$$\left\{ 1 - \left( \frac{s-1}{\nu-1} \right)^\alpha : s \in \{1, 2, \ldots, \nu - 1\} \right\}. \tag{19}$$

*Proof.* Note that there are $\nu - 1$ values given in (19) while there are $2\nu - 2$ points in the cluster, excluding $x_p$ and $x_{p^*}$. Each value in (19) is the suboptimality gap for two distinct points in $\mathcal{P} \setminus \{x_p, x_{p^*}\}$. We can construct such a dataset $\mathcal{P}$ in the following manner.

We index these points as $p, p_1, p_2, \ldots, p_{2\nu-1}$. Suppose the pairwise distance values are such that

$$d_{p,p_1} > d_{p,p_2} > \cdots > d_{p,p_{\nu-1}} > d_{p,p_\nu} =: d_{p,p^*}, \text{ and } d_{p,p_{\nu+1}} < d_{p,p_{\nu+2}} < \cdots < d_{p,p_{2\nu-1}} \text{ such that}$$
$$\forall s \in \{1, 2, \ldots, \nu-1\} \text{ we have that } d_{p,p_{\nu-s}} = d_{p,p_{\nu+s}} \implies d_{p,p_i} = d_{p,p_{2\nu-i}}. \tag{20}$$

We can then construct a $2\nu \times 2\nu$ distance matrix $D$ in the following manner. The first row of $D$ is

$$D[0,:] := [0 \quad d_{p,p_1} \quad d_{p,p_2} \quad \cdots \quad d_{p,p_{\nu-1}} \quad d_{p,p^*} \quad d_{p,p_{\nu+1}} \quad \cdots \quad d_{p,p_{2\nu-2}} \quad d_{p,p_{2\nu-1}}].$$

The $i$th row of $D$ is obtained by carrying out $i$ circular shifts on the initial row $D[0,:]$ shown above. Thus $D$ is a circulant matrix and we can see $D[i,j]$ and $D[j,i]$ to be as follows.

$$D[i,j] = \begin{cases} d_{p,p_{j-i}} & \text{if } j > i \\ d_{p,p_{2\nu-(i-j)}} & \text{if } j < i, \end{cases} \quad \text{and} \quad D[j,i] = \begin{cases} d_{p,p_{i-j}} & \text{if } i > j \\ d_{p,p_{2\nu-(j-i)}} & \text{if } i < j. \end{cases}$$

Then using (20) we have that $D[i,j] = D[j,i]$ for all $i \neq j$ and the diagonal entries are all $0$. Thus $D$ is symmetric. In addition, the distance values of the points to any point in the cluster take the same set of values. Suppose $d_{p,p^*} =: r > 0$ and $d_{p,p_1} = 2r$. Choose an $\alpha > 0$ and let

$$d_{p,p_{2\nu-i}} = d_{p,p_i} := r \left( 2 - \left( \frac{s-1}{\nu-1} \right)^\alpha \right), \forall s \in \{1, 2, \ldots, \nu - 1\}.$$

Then $D[i,j] \leq D[i,k] + D[k,j]$ for any three distinct $i, j, k$ as the sum of any two elements is greater than $2r$, which is the largest element in $D$. Thus the distance values in $D$ satisfy the triangle inequality, and $D$ is a valid distance matrix. The suboptimality gaps for any point in the cluster is $\Delta_{p,p_i} = d_{p,p_i} - d_{p,p^*} = r(1 - ((i-1)/(\nu-1))^\alpha)$, choosing $r = 1$ finishes the required construction. $\square$

### B.1.2 Proof of Theorem 4.1

*Proof.* ANNTri makes an error in finding the nearest neighbor for some point with probability $\mathbb{P}(\text{SETri is wrong for some } x_j, j \in \{1, 2, \ldots, n\})$. We show that probability is at most $n\xi = \delta$, where the confidence level $\xi$ for each execution of SETri is set to be $\delta/n$. We use induction on $s \in \mathbb{N}$ to obtain that

$$\mathbb{P}(\forall j \in \{1, 2, \ldots, s\}, k \neq j, \max\{L_{j,k}(t), L_{j,k}^\triangle(t)\} \leq d_{j,k} \leq \min\{U_{j,k}(t), U_{j,k}^\triangle(t)\}) \geq 1 - s\xi. \tag{21}$$

Consider the base case, i.e., point $x_1$. From the initialization of ANNTri 1, $\min\{U_{1,k}(t), U_{1,k}^\triangle(t)\} = U_{1,k}(t), \min\{L_{1,k}(t), L_{1,k}^\triangle(t)\} = L_{1,k}(t)$ for all $k \neq 1$. Using (3) we have $L_{1,k}(t) \leq d_{1,k} \leq U_{1,k}(t)$ with probability $1 - \delta/n$, and since $\xi$ is $\delta/n$ the base case is true. Assume the hypothesis (21) is true for some $s$. We show that it is true for $s + 1$ as well. We can bound the error event as follows.

$$\mathbb{P}(\exists j \in \{1, \ldots, s+1\}, k \neq j : d_{j,k} \notin [\max\{L_{j,k}(t), L_{j,k}^\triangle(t)\}, \min\{U_{j,k}(t), U_{j,k}^\triangle(t)\}]) \tag{22}$$

$$= \mathbb{P}(\exists j \in \{1, \ldots, s\}, k \neq j : d_{j,k} \notin [\max\{L_{j,k}(t), L_{j,k}^\triangle(t)\}, \min\{U_{j,k}(t), U_{j,k}^\triangle(t)\}])$$

$$\quad + \mathbb{P}\bigg( \{k \neq s+1 : d_{s+1,k} \notin [\max\{L_{s+1,k}(t), L_{s+1,k}^\triangle(t)\}, \min\{U_{s+1,k}(t), U_{s+1,k}^\triangle(t)\}]\}$$

$$\quad \cap \{\forall j \in \{1, 2, \ldots, s\}, k \neq j, \max\{L_{j,k}(t), L_{j,k}^\triangle(t)\} \leq d_{j,k} \leq \min\{U_{j,k}(t), U_{j,k}^\triangle(t)\}\} \bigg)$$

From (21) the first summand in the RHS of (22) is at most $s\xi$. In the event corresponding to the second term, all the bounds used by `SETri` for $d_{j,k}, j \leq s, k \neq j$ are correct. Since $U_{s+1,.}^{\triangle}$ and $L_{s+1,.}^{\triangle}$ are both deterministically obtained (see (7), (8)) from them, they are correct as well. Thus

$$\mathbb{P}(\max\{L_{s+1,k}(t), L_{s+1,k}^{\triangle}(t)\} \leq d_{s+1,k} \leq \min\{U_{s+1,k}(t), U_{s+1,k}^{\triangle}(t)\})$$
$$= \mathbb{P}(L_{s+1,k}(t) \leq d_{s+1,k} \leq U_{s+1,k}(t)) \geq 1 - \xi.$$

Hence the second summand in the RHS of (22) is at most $\xi$. This proves (21) for $s+1$ and completes the induction.

Thus with probability $1 - n\xi = 1 - \delta$, the bounds obtained by `SETri` for finding $x_{j^*}, j \in \{1, \ldots, n\}$ are all correct. We show that `ANNTri` correctly finds all nearest neighbors if the bounds are correct. For if not, suppose `SETri` returns the wrong nearest neighbor of $x_j$ which happens only if $x_{j^*}$ is not the last point in the active set. $x_{j^*} \notin \mathcal{A}$ because some other point $x_k \in \mathcal{A}$ eliminates it. Then $d_{j,k} < \min\{U_{j,k}, U_{j,k}^{\triangle}\} < \max\{L_{j,j^*}, L_{j,j^*}^{\triangle}\} < d_{j,j^*}$, which contradicts the fact that $j^*$ is the nearest neighbor. $\qquad\square$

### B.1.3 Proof of Lemma 4.3

*Proof.* Consider a point $x_i, i < j$ which satisfies the first part of (10). If $x_j \in \mathcal{A}_i(0)$ and $x_k \in \mathcal{A}_i(0)$, then neither $x_j$ and $x_k$ were eliminated without sampling when `SEEasyi` was called for $x_i$ and hence $T_{i,j} \geq 1$ and $T_{i,k} \geq 1$. Then we have that

$$4C_{\delta/n}(T_{i,j}(t_j)) + 2C_{\delta/n}(T_{i,k}(t_k)) \leq 6C_{\delta/n}(1) \leq d_{i,k} - 2d_{i,j}$$

and $x_k \notin \mathcal{A}_j(0)$ by Lemma B.3. The second part of (10) ensures that $\{x_j, x_k\} \subseteq \mathcal{A}_i(0)$ as shown next. The points eliminated from being the nearest neighbor of $x_i$ using triangle inequality are $\mathcal{A}_i(0)^{\complement} = \cup_{m<i}\{\ell : 2U_{m,i} < L_{m,\ell}\}$. If the bounds obtained by `SEEasy` for all $m < i$ are correct,

$$\{\ell : 2U_{m,i} < L_{m,\ell}\} \subseteq \{\ell : 2d_{m,i} < d_{m,\ell}\} \implies \mathcal{A}_i(0)^{\complement} \subseteq \cup_{m<i}\{\ell : 2d_{m,i} < d_{m,\ell}\}.$$

Hence if the second condition of (10) is satisfied, then $\{j, k\} \subseteq \mathcal{A}_i(0)$ and we are done. $\qquad\square$

### B.1.4 Proof of Theorem 4.4

*Proof.* Let $x_j$ be the point on which `SEEasy` is called. Consider the case $j < k$. If $\mathbb{1}_{[A_{j,k}]} = 0$ then $x_k \notin \mathcal{A}_j(0)$ and no $\mathsf{Q}(j,k)$ queries are made. Otherwise, $x_k$ can be in the active set and from (4) at most $H_{j,k}$ samples of $d_{j,k}$ are taken. Now consider the case $k < j$. Samples of $d_{j,k}$ are only queried if $x_k \in \mathcal{A}_j(0)$. If $x_j \notin \mathcal{A}_k(0)$, i.e., $x_j$ was eliminated when `SEEasy` was called for $x_k$ then no $\mathsf{Q}(k,j)$ queries made at that round. Again from (4) at most $H_{j,k}$ samples of $d_{j,k}$ are taken by `SEEasy` while finding $x_{j^*}$. If however $\mathbb{1}_{[A_{k,j}]} = 1$, then $\mathsf{Q}(k,j)$ queries were made while finding $x_{k^*}$ and let the number of those samples be $\#\mathsf{Q}(k,j)$. Because of the sampling procedure of `SEEasy`, at most $(H_{j,k} - \#\mathsf{Q}(k,j))_+$ queries are made for $d_{j,k}$. The total number of $\mathsf{Q}(j,k)$ and $\mathsf{Q}(k,j)$ queries is $\max\{H_{j,k}, \#\mathsf{Q}(k,j)\}$, and since $\#\mathsf{Q}(k,j) \leq H_{k,j}$, we get the result. $\qquad\square$

## B.2 Details for Section 4.3

In this section, we consider a case where `ANNTri` achieves complexity that scales like $O(n^{1.5})$ as well as $O(n\log(n))$, the known optimal rate for the all nearest neighbors problem for noiseless data. To do this, we first prove a lemma about the complexity of learning with clustered data. In particular, we show that if the data comes from two well separated clusters, then the complexity of learning the nearest neighbor graph can be bounded as the complexity of learning the nearest neighbors of two points looking at the full dataset and the complexity of learning the remaining nearest neighbors graphs on each of the clusters.

**Lemma B.5.** *Consider* $\mathcal{X} = \mathcal{C}_1 \cup \mathcal{C}_2$ *where* $\mathcal{C}_1$ *and* $\mathcal{C}_2$ *both satisfy 12 for all* $i, j$. *Then* `ANNEasy` *learns the nearest neighbor graph of* $\mathcal{X}$ *with probability at least* $1 - \delta$ *in at most*

$$\mathcal{O}\left(|\mathcal{C}_1| + |\mathcal{C}_2| + \mathcal{H}_{\mathcal{C}_1} + \mathcal{H}_{\mathcal{C}_2}\right) \tag{23}$$

*samples independent of the order in which it finds nearest neighbors where* $\mathcal{H}_{\mathcal{C}_i}$ *denotes the complexity of learning the nearest neighbor graph of cluster* $\mathcal{C}_i$ *as bounded by B.6.*

The above lemma implies that for the first point explored in each cluster, it is necessary to look at all other points in the dataset, but for all other points, it is only necessary to search within that point's respective cluster.

*Proof.* Choose a random order of points and fix it. Without loss of generality, we assume that $x_1 \in \mathcal{C}_1$. Let $j_2$ be the first point visited in $C_2$. Throughout, we will ignore reused samples since they only contribute at most a factor of 2 to the sample complexity as can be seen by Theorems 4.4 and B.6 and we seek an upper bound. Via standard analysis for successive elimination, $x_{1*}$ can be be found in $\mathcal{O}\left(\sum_{j=2}^n H_{1,j}\right) = |\mathcal{C}_2| + \mathcal{O}\left(\sum_{j\in\mathcal{C}_1\setminus\{x_1\}} H_{1,j}\right)$ samples with probability at least $1 - \delta/n$. For all $i = 2, \cdots, j_2 - 1$,

$$\mathcal{A}_i(0)^c \supset \{\mathcal{A}_1(0) \cap \{k : d_{i,k} \geq 6d_{i,j} - 3d_{1,1^*}\}\} \supset \{\mathcal{X}\setminus\{x_1\} \cap \mathcal{C}_2\} = \mathcal{C}_2$$

which implies that $x_{i*} \in \mathcal{C}_1$. For $x_{j_2}$ we may trivially say that $\mathcal{A}_{j_2}(0)^c \supset \{j_2\}$ so $x_{j_2^*}$ can be learned in $\mathcal{O}\left(\sum_{l\neq j_2} H_{i,j}\right) = |\mathcal{C}_1| + \mathcal{O}\left(\sum_{j\in\mathcal{C}_2\setminus\{x_{j_2}\}} H_{j_2,j}\right)$ samples with probability at least $1 - \delta/n$. We conclude by showing that for all remaining $x_i$, if $x_i \in \mathcal{C}_1$, then $\mathcal{A}_i(0) \subset \mathcal{C}_1$ and if $x_i \in \mathcal{C}_2$, then $\mathcal{A}_i(0) \subset \mathcal{C}_2$. Consider the case that $x_1 \in \mathcal{C}_1$. Suppose that $\exists x_j \in \mathcal{A}_i(0) \cap \mathcal{C}_2$. Then $2U_{1,i} > L_{1,j}$.

$$U_{i,1} = \hat{d}_{1,i} + C_{\delta/n}(T_{1,i}) \leq d_{1,i} + 2C_{\delta/n}(T_{1,i}) = d_{1,i} + 2C_{\delta/n}(1)$$

where the first inequality holds by B.2. Similarly,

$$L_{1,j} = \hat{d}_{1,j} - C_{\delta/n}(T_{1,j}) \geq d_{1,j} - 2C_{\delta/n}(T_{1,j}) \geq d_{1,j} - 2C_{\delta/n}(1)$$

Then $2(d_{1,i} + 2C_{\delta/n}(1)) \geq 2U_{1,i} > L_{1,j} \geq d_{1,j} - 2C_{\delta/n}(1) \implies d_{1,j} < 2d_{1,i} + 6C_{\delta/n}(1) \implies j \in \mathcal{C}_1$ which is a contradiction. A similar proof holds for $x_i \in \mathcal{C}_2$. It remains to argue that $j_2$ can be any number between 2 (by assumption that $x_1 \in \mathcal{C}_1$) and $|\mathcal{C}_1| + 1$ without affecting the bound on the complexity. By the assumption that $\mathcal{C}_1$ and $\mathcal{C}_2$ satisfy 12, out of cluster points can be eliminated in a single sample. Therefore, for any $j_2$, $\sum_{l\in\mathcal{C}_1} H_{j_2,l} = |\mathcal{C}_1|$. Then we have that the total complexity is $\mathcal{O}\left(|\mathcal{C}_1| + |\mathcal{C}_2| + \mathcal{H}_{\mathcal{C}_1} + \mathcal{H}_{\mathcal{C}_2}\right) \forall j_2$. Since we have considered general orders of finding each nearest neighbor, we are done. $\square$

### B.2.1 Proof of Theorem 4.5

*Proof.* By assumption, the dataset $\mathcal{X} = \cup_{i=1}^c \mathcal{C}_i$ with each cluster satisfies Equation 12. Therefore, for all $m$, $\mathcal{X} = \mathcal{C}_m \cup (\cup_{j\neq m}\mathcal{C}_i)$. By applying Lemma B.5, iteratively, we bound the complexity in terms of the the complexity of learning the nearest neighbor graph of $\mathcal{C}_m$, the complexity of learning the nearest neighbor graph of $\cup_{j\neq m}\mathcal{C}_i$, and an additive penalty of $n$ which accounts for the samples taken between the two. Since $\mathcal{X}$ is a union of $c$ clusters, this process may repeat $c$ times. Therefore the total complexity can be bounded as

$$\mathcal{O}\left(cn + \sum_{i=1}^c \sum_{j,k\in\mathcal{C}_i} H_{j,k}\right)$$

Taking $c = \sqrt{n}$, we see that the above sum is $\mathcal{O}\left(n^{1.5}\overline{\Delta^{-2}}\right)$ where $\overline{\Delta^{-2}} = \frac{1}{c*n}\sum_{i=1}^c \sum_{j,k\in\mathcal{C}_i} H_{j,k}$ is the average number of times intra-cluster distances are sampled. By contrast, the complexity for random sampling is $\mathcal{O}(n^2\Delta_{\min}^{-2})$ where $\Delta_{\min}^{-2} := \min_{j,k} H_{j,k}$. Comparing the two, we see that the latter is larger by at least a factor of $\mathcal{O}(\sqrt{n})$. $\square$

### B.2.2 Proof of Lemma 4.6

Next we use Lemma B.5 to show that for datasets such that the clusters nest, we can achieve complexity scaling in $\mathcal{O}(n\log(n)\overline{\Delta^{-2}})$. In particular, we will recursively apply Lemma B.5 to show that clusters can be broken into subclusters and initial active sets shrink in diadic splits.

*Proof.* Before we prove the theorem, we begin by introducing some notation to make this proof concise. Recall that we have assumed that $\mathcal{X}$ can be written as a hierarchy of clusters and sub clusters that form a balanced tree. We will denote the root of the tree with the full dataset as the $0^{th}$ level and

each split in that level with be indexed by $i = 1, \cdots, 2^\ell$ where $\ell = 0, \cdots, \log(n/\nu) - 1$ denotes the level. For notational ease, we take $\mathcal{C}_{0,1} \equiv \mathcal{X}$. $\mathcal{C}_{\ell,i}$ denotes the $i^{th}$ cluster at the $\ell^{th}$ level of the tree which may be split into subclusters if $\ell < \log(n/\nu) - 1$. The idea will be to traverse the tree and split clusters into subclusters while keeping track of the number of between cluster samples that were be necessary due to the bound in Lemma B.5. We let $\mathcal{H}_{\mathcal{C}_{\ell,i}}$ denote complexity of learning the nearest neighbor graph of $\mathcal{C}_{\ell,i}$.

Randomize the order and fix it. We will proceed by recursively applying Lemma B.5 to bound the complexity of learning the full nearest neighbor graph of a cluster in terms of learning it for each subcluster plus an additive penalty. By Lemma B.5 the complexity of finding the nearest neighbor graph of $\mathcal{X}$ can be upper bounded as

$$\mathcal{O}\left(|\mathcal{C}_{1,1}| + |\mathcal{C}_{1,2}| + \mathcal{H}_{\mathcal{C}_{1,1}} + \mathcal{H}_{\mathcal{C}_{1,2}}\right) = \mathcal{O}\left(n + \mathcal{H}_{\mathcal{C}_{1,1}} + \mathcal{H}_{\mathcal{C}_{1,2}}\right).$$

We may again apply Lemma B.5 to $\mathcal{C}_{1,1}$ and $\mathcal{C}_{1,2}$. to bound their complexities as $\mathcal{O}\left(\frac{n}{2} + \mathcal{H}_{\mathcal{C}_{2,1}} + \mathcal{H}_{\mathcal{C}_{2,2}}\right)$ and $\mathcal{O}\left(\frac{n}{2} + \mathcal{H}_{\mathcal{C}_{2,3}} + \mathcal{H}_{\mathcal{C}_{2,4}}\right)$ respectively where $\mathcal{C}_{1,1} = \mathcal{C}_{2,1} \cup \mathcal{C}_{2,2}$ and $\mathcal{C}_{1,2} = \mathcal{C}_{2,3} \cup \mathcal{C}_{2,4}$. Therefore, similar to the above level, the total additive penalty for samples between clusters is $n$ for the level. We may continue this process of splitting and paying the penalty of $n/2^\ell \times 2^\ell$ between cluster samples down to the bottom level $\ell = \log(n/\nu)$ with clusters of size $\nu$.

Therefore, we may write the complexity as

$$\mathcal{O}\left(n \log\left(\frac{n}{\nu}\right) + \sum_{i=1}^{n/\nu} \sum_{j,k \in \mathcal{C}_{\log(n/\nu),i}} H_{j,k}\right). \tag{24}$$

Ignoring logarithmic factors, each complexity term $H_{j,k}$ is of the order $\mathcal{O}(\Delta_{j,k}^{-2})$. Therefore the entire summation is of the order

$$\mathcal{O}\left(n \log\left(\frac{n}{\nu}\right) + n\nu\overline{\Delta^{-2}}\right)$$

where $\overline{\Delta^{-2}} := \frac{1}{n\nu} \sum_{i=1}^{n/\nu} \sum_{j,k \in \mathcal{C}_i} \log(n^2/(\delta\Delta_{j,k}))\Delta_{j,k}^{-2}$ is the average complexity. Recalling that $\nu = \mathcal{O}(\log(n))$, we are done. $\qquad\square$

### B.3 Sample complexity without using triangle inequality

**Theorem B.6.** *With probability $1 - \delta$, the number of oracle queries made by `ANNTri` and `ANNEasy` if all triangle bounds are ignored is at most*

$$\mathcal{O}\left(\sum_{i<j} \max\left\{\frac{\log(n^2/(\delta\Delta_{i,j}))}{\Delta_{i,j}^2}, \frac{\log(n^2/(\delta\Delta_{j,i}))}{\Delta_{j,i}^2}\right\}\right). \tag{25}$$

In the experiments, the process of using `ANNTri` and ignoring triangle inequality bounds is referred to as `ANN`.

*Proof.* In the case that triangle bounds are ignored, `ANNTri` and `ANNEasy` are the same. Consider the $i^{th}$ round where we seek to identify $x_{i^*}$ with probability $1 - \delta/n$. `ANNTri` has found $x_{\ell^*}$ for all $\ell < i$, in particular, it has evaluated $\hat{d}_{\ell,i}, U_{\ell,i}, L_{\ell,i}$. For every $x_j \neq x_{i^*}, x_j \in \mathcal{A}_i(0)$, we can bound the number of $Q(i,j)$ queries in the following manner. Suppose $j > i$ and $i^* > i$, so that at the beginning of the $i^{th}$ round we have that $T_{i,j} = T_{i,i^*} = 0$. From (3), with probability $1 - \delta/n$, $x_{i^*}$ is the last point in the active set. The point $x_j$ is eliminated from the active set at time $t_j$ if the following is true.

$$U_{i,i^*}(t_j) \overset{(a)}{\leq} d_{i,i^*} + 2C_{\delta/n}(T_{i^*}(t_j)) < d_{i,j} - 2C_{\delta/n}(T_j(t_j)) \overset{(b)}{\leq} L_{i,j}(t_j),$$
$$\implies 4C_{\delta/n}(t_j) < d_{i,j} - d_{i,i^*} = \Delta_{i,j}. \tag{26}$$

Inequalities (a), (b) are due to Lemma B.2, and the fact that if $j$ is eliminated at time $t_j$, then $T_{i,j}(t_j) = t_j$. From the property of the $C_{\delta/n}(\cdot)$ function, (26) is ensured when the number of samples of $d_{i,j}$ is

$$t_j \leq \left\lceil \kappa \frac{\log(n^2/(\delta\Delta_{i,j}/4))}{(\Delta_{i,j}/4)^2} \right\rceil.$$

We now consider the cases when at least one of $i^*, j$ are less than $i$.

$i^* > i, j < i$: In this case, at the beginning of the $i^{th}$ round $T_{i,j}$ is equal to the number of $\mathsf{Q}(j,i)$ queries made (denoted as $\#\mathsf{Q}(j,i)$) while finding $x_{j^*}$:

$$\#\mathsf{Q}(j,i) \le \left\lceil \kappa \frac{\log(n^2/(\delta \Delta_{j,i}/4))}{(\Delta_{j,i}/4)^2} \right\rceil.$$

If $\#\mathsf{Q}(j,i) > t_j$, then no further $\mathsf{Q}(i,j)$ queries are made in the $i^{th}$ round, as argued next. Because the sampling procedure of $\mathtt{SETri}$ queries all points who have the minimum number of samples at current time, if a query $\mathsf{Q}(i,j)$ is made at time $t+1$, that implies $T_{i,i^*}(t) = \#\mathsf{Q}(j,i)$. But then $j$ is not in the active set at time $t$ as

$$U_{i,i^*}(\#\mathsf{Q}(j,i)) < U_{i,i^*}(t_j) < d_{i,j} - 2C_{\delta/n}(t_j) < d_{i,j} - 2C_{\delta/n}(\#\mathsf{Q}(j,i)) = L_{i,j}(\#\mathsf{Q}(j,i))$$

and hence $\mathsf{Q}(i,j)$ is not made. If $\#\mathsf{Q}(j,i) < t_j$, then $x_j$ is eliminated when $t_j - \#\mathsf{Q}(j,i)$ more samples of $d_{i,j}$ have been queried. Thus the total number of samples of $d_{i,j}$ is at most $\max\{t_j, \#\mathsf{Q}(j,i)\}$.

The other two cases of 1) $i^* < i, j > i$, and 2) $i^* < i, j < i$ can be handled similarly. $\qquad\square$

## C  Average case performance of $\mathtt{ANNEasy}$

We can obtain a different expression for the number of oracle queries if all the random quantities during a run of the algorithm take their expected values. In particular, Lemma 4.3 can be relaxed to the following.

**Lemma C.1.** *If all bounds obtained by $\mathtt{SEEasy}$ are correct and all the random quantities take their expected values, then for some $i < j$ such that $x_j \ne x_{i^*} \ne x_k$ if we have that*

$$d_{i,k} > 6d_{i,j} - 3d_{i,i^*}, \quad and \quad \{j,k\} \cap (\cup_{m<i}\{\ell : 2d_{m,i} < d_{m,\ell}\}) = \emptyset, \tag{27}$$

*then $2U_{i,j} < L_{i,k}$ and hence $x_k \notin \mathcal{A}_j(0)$.*

*Proof.* In the good event, the point $x_{i^*}$ is the last element in the active set $\mathcal{A}_i$ and points $x_j, x_k$ have been eliminated from $\mathcal{A}_i$ at some prior times $t_j, t_k$ respectively. Both $t_j > 0$ and $t_k > 0$ as $\{x_j, x_k\} \subset \mathcal{A}_i(0)$ is ensured by the second part of the condition, as shown in the proof of Lemma 4.3. At time $t_j$, we have that

$$\min_\ell \hat{d}_{i,\ell} + C_{\delta/n}(t_j) \le \min_\ell U_{i,\ell} \le L_{i,j} \le \hat{d}_{i,j} - C_{\delta/n}(t_j). \tag{28}$$

If all the random quantities take their expected values, then $\hat{d}_{i,\ell} = d_{i,\ell} \forall \ell \ne i$ and we have that

$$d_{i,i^*} + C_{\delta/n}(t_j) \le d_{i,j} - C_{\delta/n}(t_j) \implies C_{\delta/n}(t_j) \le \Delta_{i,j}/2. \tag{29}$$

Under the assumption, $\hat{d}_{i,j} = d_{i,j}$ and using the definition of its upper and lower confidence bounds, we get that $\mathbb{E}[L_{i,j}] \ge d_{i,j} - \Delta_{i,j}/2$ and $\mathbb{E}[U_{i,j}] \le d_{i,j} + \Delta_{i,j}/2$. Similar bounds are true for $x_k$. Then

$$d_{i,k} > 6d_{i,j} - 3d_{i,i^*} \implies d_{i,k} - \frac{d_{i,k} - d_{i,i^*}}{2} d_{i,k} - \frac{\Delta_{i,k}}{2}$$
$$> 2\left(d_{i,j} + \frac{d_{i,j} - d_{i,i^*}}{2}\right) = 2\left(d_{i,j} + \frac{\Delta_{i,j}}{2}\right),$$

which implies that $L_{i,k} = \mathbb{E}[L_{i,k}] > 2\mathbb{E}[U_{i,j}] = 2U_{i,j}$ and $x_k \notin \mathcal{A}_j(0)$. $\qquad\square$

If all the random quantities take their expected value, then using Lemma C.1 and the elimination criterion of $\mathtt{ANNEasy}$ (Lemma 4.2), the complement of the initial active set $\mathcal{A}_j(0)$ (also called the elimination set) can be characterized in the following manner.

$$\mathcal{A}_j(0)^{\complement} = \cup_{i<j:j\in\mathcal{A}_i(0)}\{\mathcal{A}_i(0) \cap \{k : 2U_{i,j} < L_{i,k}\}\}$$
$$\supseteq \cup_{i<j:j\in\mathcal{A}_i(0)}\{\mathcal{A}_i(0) \cap \{k : d_{i,k} > 6d_{i,j} - 3d_{i,i^*}\}\}. \tag{30}$$

Replacing the indicator $\mathbb{1}_{[A_{j,k}]}$ in Theorem 4.4 with an indicator for the non-membership of point $x_k$ in the set (30) gives us an upper bound to the sample complexity of `ANNEasy` that is valid when all random quantities take their expected values.

To gain an idea of the savings achieved by our algorithm in comparison to the random sampling, we evaluate the sample complexity expressions for an example dataset. The dataset we look at consists of $c$ clusters, each cluster containing $n/c > 1$ points. The points are indexed such that the $m$th cluster is $\mathcal{C}_m := \{x_{\underline{m}}, x_{1+\underline{m}}, \ldots, x_{\overline{m}}\}$, where

$$\underline{m} := 1 + (m-1)n/c \tag{31}$$
$$\text{and}$$
$$\overline{m} := mn/c \tag{32}$$

for all $m \in [c]$. Suppose the distances between the points are such that for any pair $\{x_i, x_j\} \subseteq \mathcal{C}_m$, the set of points

$$\{x_k : d_{i,k} < 6d_{i,j} - 3d_{i,i^*}\} \subseteq \mathcal{C}_m. \tag{33}$$

The above condition is ensured if the smallest distance between two points belonging to different clusters is at least six times the diameter of any cluster.

**Lemma C.2.** *Consider a dataset which satisfies the condition in* (33). *If all random quantities take their expected values,* `ANNEasy` *uses $O(\sqrt{n})$ fewer oracle queries than the random sampling baseline to learn the nearest neighbor graph.*

*Proof.* In the following we assume that all random quantities take their expected values. We can find the points that are definitely eliminated using the triangle inequality when `ANNEasy` is called using (30). The elimination set $\mathcal{A}_1(0)^\complement = \{x_1\}$. For a point $x_i \in \mathcal{C}_1 \notin \mathcal{E}_1$, from (30), (33) we get that

$$\mathcal{A}_i(0)^\complement \supseteq \{\mathcal{A}_1(0) \cap \{k : d_{1,k} > 6d_{1,i} - 3d_{1,1^*}\}\} \supseteq \{(\mathcal{X} \setminus \{x_1\}) \cap \mathcal{C}_1^\complement\} = \mathcal{C}_1^\complement.$$

Thus $\mathcal{A}_i(0) \subseteq \mathcal{C}_1$ for all $x_i \in \mathcal{C}_1$. Point $x_{\underline{m}}$ is the first point processed by `ANNEasy` in the $m$th cluster. Suppose there exists a point $x_j \in \mathcal{C}_m \cap \mathcal{A}_{\underline{m}}(0)^\complement$, we show next that leads to a contradiction. Since $x_j \notin \mathcal{A}_{\underline{m}}(0)$, there is a point $x_i \in \mathcal{C}_{m'}$ with $i < j, m' < m$ such that $2U_{i,\underline{m}} < L_{i,j}$. Let $\text{Diam}(\mathcal{C}_m) := \max_{x_\ell, x_k \in \mathcal{C}_m} d_{\ell,k}$ be the diameter of cluster $\mathcal{C}_m$ (similarly for $\mathcal{C}_{m'}$) and let $D(\mathcal{C}_{m'}, \mathcal{C}_m) := \min_{x_\ell \in \mathcal{C}_{m'}, x_k \in \mathcal{C}_m} d_{\ell,k}$ be the minimum inter-cluster distance. Since the random quantities take their expected values, we have that

$$U_{i,\underline{m}} \geq d_{i,\underline{m}} + \frac{d_{i,\underline{m}} - d_{i,i^*}}{2} \implies 2U_{i,\underline{m}} \geq 3D(\mathcal{C}_{m'}, \mathcal{C}_m) - \text{Diam}(\mathcal{C}_{m'}),$$

$$L_{i,j} \leq d_{i,j} - \frac{d_{i,j} - d_{i,i^*}}{2} \implies L_{i,j} \leq \frac{\text{Diam}(\mathcal{C}_{m'}) + D(\mathcal{C}_{m'}, \mathcal{C}_m) + \text{Diam}(\mathcal{C}_m)}{2} + \frac{\text{Diam}(\mathcal{C}_{m'})}{2}.$$

Using $2U_{i,\underline{m}} < L_{i,j}$ with the above inequalities implies that $2.5D(\mathcal{C}_{m'}, \mathcal{C}_m) < 2D(\mathcal{C}_{m'}) + 0.5D(\mathcal{C}_m)$, which is a contradiction as from (33) we have that $D(\mathcal{C}_{m'}, \mathcal{C}_m) \geq \max\{3D(\mathcal{C}_{m'}), 3D(\mathcal{C}_m)\}$. Thus we have that $\mathcal{C}_m \cap \mathcal{A}_{\underline{m}}(0)^\complement = \emptyset$. For any $x_j \in \mathcal{C}_m, j \neq \underline{m}$ we have that $x_j \in \mathcal{A}_{\underline{m}}(0)$ and hence from (30),

$$\mathcal{A}_j(0)^\complement \supseteq \{\mathcal{A}_{\underline{m}}(0) \cap \{k : d_{\underline{m},k} > 6d_{\underline{m},j} - 3d_{\underline{m},\underline{m}^*}\}\} \supseteq \mathcal{C}_m^\complement.$$

Based on the above discussion, we have a lower bound on the number of points present in the elimination set $\mathcal{A}_j(0)^\complement$ for any $x_j \in \mathcal{C}_m$. By choosing the following values for the indicator in (25)

$$\mathbb{1}_{[A_{j,k}]} = \begin{cases} 0 & \text{if } x_j \in \mathcal{C}_m \setminus \{x_{\underline{m}}\} \text{ and } x_k \notin \mathcal{C}_m, \\ 1 & \text{otherwise,} \end{cases}$$

we get the following upper bound to the number of oracle queries, where $x_{\overline{m}}$ is the last point in $\mathcal{C}_m$.

$$\mathcal{O}\left( \sum_{m=1}^{c} \left( \sum_{k > \underline{m}} H_{\underline{m},k} + \sum_{k < \underline{m}} H_{\underline{m},k} - \sum_{\ell=1}^{m-1} H_{\underline{m},\ell} + \sum_{\ell=1}^{m-1} (H_{\underline{m},\ell} - H_{\ell,\underline{m}})_+ \right. \right.$$
$$\left. \left. + \sum_{p > \underline{m}}^{\overline{m}} \sum_{q > p}^{\overline{m}} \max\{H_{p,q}, H_{q,p}\} \right) \right) \tag{34}$$

where $\underline{m}$ and $\overline{m}$ are defined in (31) and are functions of $m$. The number of terms in the sum above is $\mathcal{O}(cn + (n/c)^2)$. A uniform sampling baseline approach would have $\mathcal{O}(n^2)$ terms in its sample complexity. Letting $c = \sqrt{n}$ gives our result. $\qquad \square$

The above lemma ensures that we have $\mathcal{O}(\sqrt{n})$ fewer terms in the sample complexity expression for `ANNEasy` compared to random sampling if the dataset satisfies (33). We can get a more precise characterization of the savings in query complexity in terms of the $\Delta_{p,q}$ values. For instance, using a single-parameter model for the distribution of $\Delta_{p,q}$ as done in [19], we can directly use their Corollary 1 in our context.

**Lemma C.3.** *Consider a clustered dataset $\mathcal{X} = \cup_{m=1}^{c}\mathcal{C}_m$ whose points satisfy (33). Each cluster contains an even number $2\nu := n/c$ of points. For any $m \in [c]$ and $x_j \in \mathcal{C}_m$, suppose the suboptimality gaps $\Delta_{j,k}$ for all $x_k \in \mathcal{C}_m$ take one of the following values, parametrized by an $\alpha > 0$:*

$$1 - \left(\frac{s-1}{\nu-1}\right)^{\alpha}, \qquad \text{where} \qquad s \in \{1,2,\dots,\nu-1\}. \tag{35}$$

*Note that there are $\nu - 1$ values given in (35) while there are $2\nu - 2$ points in the cluster, excluding $x_j$ and $x_{j^*}$. Each value in (35) is the suboptimality gap for two distinct points in $\mathcal{C}_m$. Ignoring $\log$-factors, if $\alpha = 1$ `ANNTri` finds all nearest neighbors with probability $1 - \delta$ in $O(n(\nu^2 + n))$ calls to the oracle, while uniform sampling requires $O(n^2\nu^2)$ calls for the same guarantee.*

*Proof.* By putting the clusters far from each other, one can see that there exist $\mathcal{X} = \cup_{m=1}^{c}\mathcal{C}_m$ whose points satisfy (33). Lemma B.4 shows by explicit construction that the condition on the suboptimality gaps within each cluster as stated in (35) can also be satisfied. Note that (35) is the same parametrization as equation 3 in [19].

Consider the points in the $m$th cluster, i.e., points $x_{\underline{m}}$ through $x_{\overline{m}}$. The elimination set $\mathcal{A}_{\underline{m}}(0)^{\complement}$ can be the singleton $\{x_{\underline{m}}\}$, but by Lemma C.1 for all $x_p \in \mathcal{C}_m \setminus \{x_{\underline{m}}\}, \mathcal{A}_p(0)^{\complement} \supseteq \mathcal{C}_m^{\complement}$. Finding $x_{p^*}$ is a best arm identification problem among points within the cluster $\mathcal{C}_m$. The last term in (34) counts the total number of oracle queries made by `ANNEasy` to identify the nearest neighbors of all $x_p \in \mathcal{C}_m \setminus \{x_{\underline{m}}\}$. Thus the number of oracle queries made by `ANNEasy` for identifying $x_{p^*}$ is at most $\sum_{q \neq p} H_{p,q}$, while uniform sampling will make $nH_{p,p'}$ queries, where $p' := \arg\min_{q \neq p^*} \Delta_{p,q}$.

Ignoring $\log$-factors, the sample complexity for finding $x_{p^*}$ for an $x_p \in \mathcal{C}_m$ by `ANNEasy` is

$$\tilde{\mathcal{O}}\left(\sum_{i=1}^{2\nu-1}\Delta_{p,p_i}^{-2} + \sum_{x_\ell \notin \mathcal{C}_m}\Delta_{p,\ell}^{-2}\right) = \tilde{\mathcal{O}}\left(\sum_{i=1}^{2\nu-1}\Delta_{p,p_i}^{-2} + n - 2\nu\right).$$

Corollary 1 of [19] lists the value of that sum for different choices of $\alpha$, for e.g., if $\alpha = 1$ then the sample complexity is $\tilde{\mathcal{O}}(\nu^2 + n - 2\nu)$. On the other hand, for finding $x_{p^*}$ uniform sampling would make $\tilde{\mathcal{O}}(n\Delta_{p,p'}^{-2})$, i.e., $\tilde{\mathcal{O}}(n(\nu-1)^2)$ queries. By construction of the dataset, finding the nearest neighbor of each point in $\mathcal{X}$ is equally hard. Thus `ANNTri` would make $\tilde{\mathcal{O}}(n(\nu^2 + n - 2\nu))$ queries while uniform sampling would take $\tilde{\mathcal{O}}(n^2\nu^2)$ queries. $\qquad \square$

Note that our problem setting is inherently different from the noiseless setting where all $x_{i^*}$'s can trivially be learned in $\binom{n}{2}$ samples. Due to the presence of noise in our queries, many distances must be repeatedly queried so $\binom{n}{2}$ samples is insufficient.