[Reviews · NeurIPS 2019]

Reviewer 1



After reading the author feedback, my review stays the same and in particular, I still think this is a good submission. Thanks to the authors for the thorough response! I realized that there was an additional point of discussion I didn't mention in my original review that could be helpful to discuss: you benchmark against standard Euclidean embeddings, but what about hyperbolic embeddings? Hyperbolic embeddings could encode trees and I suspect that they can handle hierarchical clustering structure like what is analyzed in Theorem 4.6. This could possibly be an interesting direction of future research. * * * * * Original review below * * * * * I found the paper very clear and easy to follow. The key ideas for limiting the number of queries (symmetry in Q(i,j) and Q(j,i), triangle inequality, concentration inequalities) are quite straightforward, and at some level much of the paper reads off as doing diligent bookkeeping with the lower and upper bounds in updating the active sets as to keep the query cost low while maintaining accuracy. My comments are minor: 1. Perhaps some discussion on whether the theory could be extended to recovering k-NN graphs would be helpful, along with what the technical hurdles are in doing so. It seems like we could use a best-k-arm strategy per node. 2. In Algorithm 1's "Require" line, there is a reference to Algorithm 2 that should probably say something like "SETri (Alg. 2)" or "SETri (Algorithm 2)" rather than just "SETri, 2". There's a similar issue in Algorithm 2's "Require" line in referencing Algorithm 1.

Reviewer 2



---------- I read the author response and am satisfied with their promise to fix the minor issues and provide a more elaborate related work. I think the Kliendesener and Luxburg (JMLR 2017) works with noisy triplets to build kNN graph. I do not feel the need to change my review score ---- The algorithm though yields good results is relatively straight-forward involving standard probability concentration bounds and using triangular inequality to the full extent. I believe the method is technically correct and the overall quality of the presentation is good. There are many minor typos and grammatical mistakes (such as equation (5), repetitions on line 181 and 183, etc.) I think the second item (Line 79) is not stated correctly, but it does not affect the algorithm or its subsequent analysis. The notation of the paper can be (and should be) simplified, it is overly technical and notations are introduced on the go within the statements (such as in Lemma 3.1 with =:). My main reservations on the quality and significance of work are its limitation to the metric case, the number of query analysis of the simplified algorithm only, the very large space complexity. In the experimental section, the distance measure defined on the real dataset is not a metric (as acknowledged by the authors), but its effect on the quality of results is not discussed. This is particularly important since triangle inequality is the fundamental building block of the algorithm. The related work section is very poorly written; very closely related work must be elaborated on. The results should be compared with techniques that build nearest neighbor graphs based on triplets (directly) without ordinal embedding. The reproducibility checklist shows that a link to the code is provided, while in the paper the author(s) state(s) that it will be provided upon publication.

Reviewer 3



The paper is clean. As a non theory person I was able to follow the motivation, the problem set-up and the main result. The intuition that not all queries Q(j,k) needs to be issued conditioned on the past queries due to the fact that we're in a metric space. The bounds looks fine, mostly it is a construction of the confidence in the estimation of Q(j,k), again, under the constraint that Q(j,k) is a metric space with metric-ish properties. As a piece of technical achievement this paper is just fine. But it can improve in the following sense of story-telling and organisation: 1) we're doing an optimal query problem, namely, querying a noisey oracle Q(j,k) to construct a nearest-neighbor graph G(x_i,x_j). Given this is the setting, I was hoping to see some kind of quantification over the entropy on all the possible nn-graph G, and how the next-best-query you're querying is maximally reducing that entropy (i.e. with this new query, I gain the most information on distinguishing valid nnGraphs from incompatible graphs) 2) It is unclear to me then, which criteria you are using to select the next query Q(i,j) to make. Are you selecting a query that maximally reduce the entropy on the space of possible nearest-neighbor graphs? Or you're selecting the query that maximally reduce the entropy on the pair-wise distance metric Q(i,j) itself? These two are different objects and you'd expect the querying scheme to be different. No doubt this distinction is somewhere in the paper, but it would be good to have it in english form, stated up-front, so a more "casual" reader, tasked with implementing this algorithm could follow. I was hoping to see a sentence like "We're try to maximally reduce the uncertainty on the space of all possible nn graphs. thus, based on our past K observations Q(j,k)_1 \dots Q(j,k)_K, we compute for all ?? the confidence bound of ?? and query the most ?? pair from the oracle, maximally reducing the uncertainty". 3) it might be good to have some notion of sub-modularity argument. From what I can read this paper uses a bandit-like approach, which is in a sense greedy, picking the most ?? query at each step instead of planning ahead a sequence of K queries that, maybe themselves do not lead to good information gain, but in conjunction leads to huge information gain. Greedy solutions are just fine if your problem is sub-modular, in this case, the entropy gain over the space of nearest-neighbor graphs is sub-modular with respect to the set of queries you are issuing, it might be good to prove this, and then you can easily justify your greedy strategy as compatible to optimal. So rating is as follows: Originality: fair. It appears to me that there has not been a paper previously that only assume general metric and noisy reads in the context of inferring nn-graph Quality: good. Math is clean, theorem is well-written, evaluation is well constructed (although it doesn't really add much given this is a theory paper anyways) Clarity: poor. No idea what the selection criteria is, and no intuition on why this particular selection criteria is optimal (or greedy-optimal) Significance: fair. Pretty clean problem, I'm sure someone would find use of it later down the line.

[Author Response · NeurIPS 2019]

**Response to Reviewer #1** We thank the reviewer for taking the time to review our submission and for their helpful suggestions. Regarding recovery of $k$-NN graphs, you are correct that a best-$k$ strategy could be used to identify individual edges. The main challenge would be working out the condition for when the triangle inequality implies elimination. A point is eliminated if we can certify that there are $k$ closer points due to either triangle inequalities or symmetry. That is ensured if for a given $j$, there are $k$ distinct previous points $i$ that satisfy the condition of Lemma 4.3. In the case of clustered data, as long as any cluster has at least $k$ points, the same separation condition will apply, and this should give an additional factor of $k$ in theorems 4.5 and 4.6. We will add a discussion of extending to $k$-NN graphs to the paper and correct the references to the algorithms.

**Response to Reviewer #2** We thank the reviewer for their comments on our work and all helpful suggestions for how to improve it. To the best of our knowledge, we are unaware of any prior work that uses noisy triplet queries to learn the NN-graph. In the noiseless setting, there are techniques for nearest neighbor search using triplets [1,2], which potentially could be modified and used as a sub-routine by an algorithm that finds the NN-graph when there is no noise. We will elaborate on works relevant to the noiseless triplet setting in the related work section. Regarding provided code, we included example code zipped in the supplementary to preserve double blindness and apologize for the confusion with the checklist. On the note of the metric assumption, in the supplementary, we compare against an implementation that does not use the triangle inequality. Compared to the method that uses triangle inequality, we see slightly worse initial performance but similar gains over random sampling at higher accuracy levels. We will move this discussion to the main body to highlight it. Thank you also for pointing out the typos. We will correct them, simplify the notation, and define the quantities in Lemma 3.1 before the statement of the Lemma.

**Response to Reviewer #3** We thank the reviewer for taking the time to review our work and all their suggestions, especially about how to clarify and contextualize the selection criteria.

*Re: quantification of entropy over NN-graphs.* That is an interesting way of looking at the problem. In some problem instances, the number of queries made by our algorithm is within a constant factor of the minimum bits required to specify the answer from a list of all possible NN-graphs. There are a total of $n^{n-1}$ possible NN-graphs, hence each can be specified using $(n-1)\log n$ bits. If the dataset consists of hierarchical clusters as in the condition for Theorem 4.6, then `ANNEasy` finds the NN-graph after making $O(n\log(n)\overline{\Delta^{-2}})$ queries. Thus even though our noisy distance oracle is weaker than an oracle that answers arbitrary yes/no queries (e.g. membership queries of the form: is the true NN-graph present in a particular subset of all possible NN-graphs?), we are able to identify the NN-graph within a factor of the optimum number of queries and a multiplicative penalty to account for the noise in each answer.

The effect of having a weaker oracle can be seen in a different problem instance where $\Omega(n^2)$ distance queries are necessary to identify the NN-graph by any algorithm that uses the weaker oracle. For example, consider a dataset consisting of points $x_i = e_i + \epsilon \in \mathbb{R}^n \ \forall i$ where $e_i$ is the unit vector with 1 in the $i$th component and 0 elsewhere, and $\epsilon$ is small independent zero-mean noise. Then all points are roughly at the same distance from each other, and for finding $x_{i*}$ we have to query $\mathsf{Q}(i,j)$ for all $j \neq i$. This is made more explicit in the discussion following Theorem 4.4.

*Re: selection criteria for $\mathsf{Q}(i,j)$.* Our algorithm iterates over points $x_i$ in the dataset and finds $x_{i*}$ before starting the procedure for $x_{i+1}$. In the $i$th round, we use a modified successive elimination algorithm for bandit best-arm identification to find $x_{i*}$. It is known that this algorithm matches instance-dependent lower bounds for best-arm identification within $\log$-factors [3]. In that sense, for a given $x_i$, our algorithm optimally selects $x_j$ while querying $\mathsf{Q}(i,j)$ to find $x_{i*}$. We will add this discussion before our reference to Algorithm 2. The triangle inequality bounds used for elimination are also optimal. [4] show that this Floyd-Warshall style approach yields the tightest upper and lower bounds on the distance matrix in the entry-wise $L_1$ norm.

The order in which the points $\{x_i\}$ are processed follows their subscript index, which is randomly chosen and fixed before starting the algorithm. Different orders in which $\{x_i\}$ are processed can affect the query complexity of our algorithm as discussed in the paragraph after Theorem 4.4. However it is not always possible to find an optimal order for $\{x_i\}$ from only noisy distance samples without assumptions on the metric. For example, if the oracle is noiseless, there are datasets where the pair $(i,j)$ with the smallest $d_{i,j}$, must be queried within the first $n$ queries to identify the NN-graph using the minimum number of queries. Since that cannot be ensured by an algorithm that only has access to information via a distance oracle, it is not possible to achieve the minimum number of queries in such examples.

[1] Haghiri, S., Ghoshdastidar, D., Luxburg, U.v.. (2017). *Comparison-Based Nearest Neighbor Search.* Proceedings of the 20th International Conference on Artificial Intelligence and Statistics, in PMLR 54:851-859

[2] Houle, Michael E., and Michael Nett (2013), *Rank cover trees for nearest neighbor search*, International Conference on Similarity Search and Applications. Springer, Berlin, Heidelberg.

[3] Emilie Kaufmann, Olivier Cappé, and Aurélien Garivier. 2016. *On the complexity of best-arm identification in multi-armed bandit models.* J. Mach. Learn. Res. 17, 1 (January 2016), 1-42.

[4] Singla, Adish, Sebastian Tschiatschek, and Andreas Krause. *Actively learning hemimetrics with applications to eliciting user preferences.* International Conference on Machine Learning. 2016.


[Meta-Review · NeurIPS 2019]

This paper defines a new learning problem of learning 1-NN graph on a metric space consisting of n points using a noisy distance oracle. Ignoring the metric structure, the problem can be viewed n separate best arm identification problems (for each node find its nearest neighbor by calling the oracle). The best arm identification problem can solved by UCB-like methods that maintain confidence intervals for each distance and has been studied extensively in the literature. The paper proposes an improved algorithm that exploits the triangle inequality and symmetry of the underlying metric in order to improve the confidence intervals. As the paper shows that the improvement doesn't help much for the worst-case metric space. However, the paper also shows that the improvement helps when the metric space consists of clusters. The paper is nicely written. The theoretical analysis is correct and non-trivial. The paper has some simple experiments. The problem definition is novel and it seems that it has some practical applications. The improvements are simple, yet non-trivial.